# Hippocampal and prefrontal processing of network topology to simulate the future

Amir-Homayoun Javadi[1,*], Beatrix Emo[2,5,*], Lorelei R. Howard[3], Fiona E. Zisch[4,5], Yichao Yu[6], Rebecca Knight[7], Joao Pinelo Silva[8] & Hugo J. Spiers[4]

Topological networks lie at the heart of our cities and social milieu. However, it remains unclear how and when the brain processes topological structures to guide future behaviour during everyday life. Using fMRI in humans and a simulation of London (UK), here we show that, specifically when new streets are entered during navigation of the city, right posterior hippocampal activity indexes the change in the number of local topological connections available for future travel and right anterior hippocampal activity reflects global properties of the street entered. When forced detours require re-planning of the route to the goal, bilateral inferior lateral prefrontal activity scales with the planning demands of a breadth-first search of future paths. These results help shape models of how hippocampal and prefrontal regions support navigation, planning and future simulation.

[1] School of Psychology, University of Kent, Canterbury CT2 7NP, UK. [2] Chair of Cognitive Science, ETH Zurich 8092, Switzerland. [3] Aging and Cognition Research Group, German Center for Neurodegenerative Diseases (DZNE), Magdeburg 39120, Germany. [4] Division of Psychology and Language Sciences, Department of Experimental Psychology, UCL Institute of Behavioural Neuroscience, University College London, London WC1H 0AP, UK. [5] Bartlett School of Architecture and Design, University College London, London WC1H 0QB, UK. [6] UCL Centre for Advanced Biomedical Imaging, University College London, London WC1E 6DD, UK. [7] School of Psychology, University of Hertfordshire, Hertfordshire AL10 9AB, UK. [8] Department of Architecture and Interior Design, University of Bahrain 840, Kingdom of Bahrain. * These authors contributed equally to this work. Correspondence and requests for materials should be addressed to H.J.S. (email: h.spiers@ucl.ac.uk).

Evidence from neuropsychology, neuroimaging and electro-physiology indicates that the hippocampus supports retrieval of the past to simulate the future[1–5]. However, prior results have mainly come from tasks requiring cued mental simulation of the future. Thus, the conditions under which the hippocampus might naturally represent information needed for the future during continuous interaction with an environment remain unknown. One candidate moment is the transition between episodes, when new options for action arise.

For all motile animals one transition is universally important: crossing spatial boundaries. When we enter a new territory, future possible paths become available, which are defined by the topology of the environment. Recent evidence from rodents has shown that the connections between spaces are over-represented by the spatial localized firing of hippocampal place cells[6], and it has been argued that hippocampal place cells may preferentially code the topology of an environment rather than its geometry[7]. During 'off-line' hippocampal replay events, when hippocampal place cells show re-activation of spatial sequences, the topological structure of an environment may be re-capitulated[8]. Such simulation of the topological structure of the environment would be useful during active navigation; however, so far little evidence for 'online' representation of topological network properties of an environment has been observed.

While the hippocampus is thought to support retrieval of memory representations to simulate future possibilities, the role of evaluating possible future states for action is argued to be the preserve of the prefrontal cortex (PFC). This is based on evidence that damage to the PFC specifically impairs planning and problem solving[9,10]. However, it is not currently clear which regions of the PFC evaluate future paths or whether information contained in topological structures is searched to support navigation. We have recently proposed that the lateral frontopolar PFC is a suitable candidate region[10]. The mechanism by which path evaluation may occur is not known. One potential mechanism is a 'tree-search' of all the future branching choices in the network. Consistent with this, recent evidence indicates that humans plan their decisions based on evaluation of each level of the decision tree before proceeding to the next level[11,12]. For a street network this would involve searching retrieved representations of all the possible path streets just beyond the next junction. Such a search mechanism is known as a breadth-first search (BFS)[13], which steps through the sequences of possible future choices one level of the decision tree at a time. Prior evidence suggests that humans may use this mechanism when planning routes from cartographic maps[14].

Here we tested the hypotheses that the hippocampus retrieves representations of the topological structure of the environment when new paths are entered in order to support goal-directed navigation and the lateral PFC performs path-planning via a BFS mechanism. We combined a graph-theoretic analysis of the city streets of London with functional magnetic resonance imaging (fMRI) data collected from participants navigating a film simulation of London's streets. Our analysis reveals that the right posterior hippocampus specifically tracks the changes in the local connections in the street network, the right anterior hippocampus tracks changes in the global properties of the streets and the bilateral lateral prefrontal activity scales with the demands of a BFS. These responses were only present when long-term memory of the environment was required to guide navigation.

## Results

**Experimental design**. To test our hypotheses, we computed graph-theoretic measures of each street segment (Fig. 1) in London's (UK) Soho region and used these to interrogate fMRI

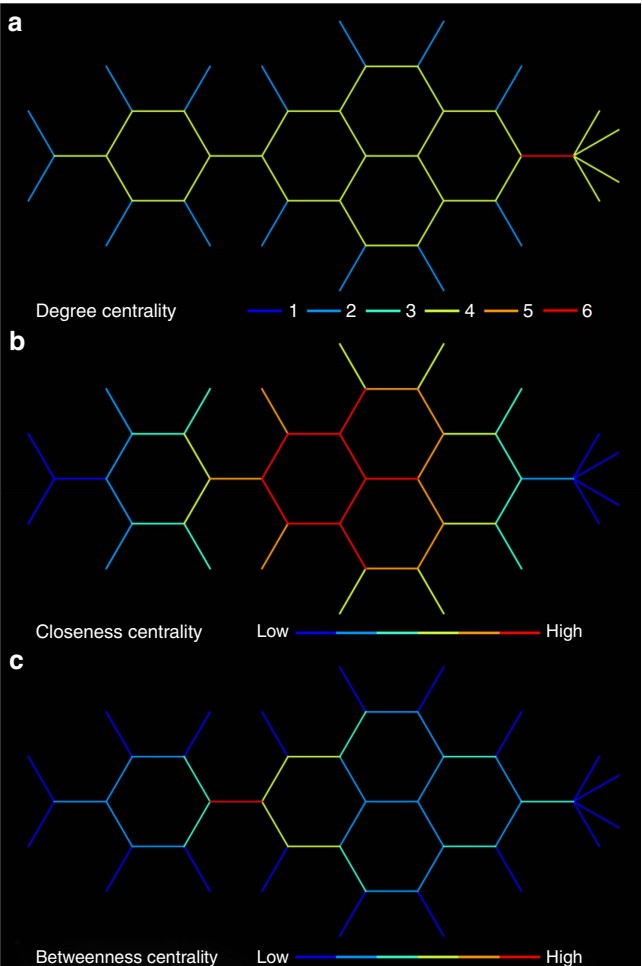

**Figure 1 | Illustration of the three centrality measures in a sample network.** The network was chosen to illustrate how the three measures of centrality record different properties of the network. Note each measure identifies different streets as having the highest value. (**a**) The highest degree centrality street reflects the fact that this street has six streets connected to it. (**b**) The highest closeness centrality streets reflect the fact that these streets are topologically closest to all other streets in the network. (**c**) The highest betweenness centrality street indicates that this street would be travelled most frequently when travelling from any one street to another.

data collected from participants navigating through a film simulation of Soho (Fig. 2 (refs 15–17) and Methods). One day after extensive *in situ* training (see Methods), the participants were scanned while watching 10 first-person-view movies of novel routes through Soho. Five movies required participants to make navigational decisions (Navigation routes), while the other five did not (Control routes). At the start of each Navigation route, participants were oriented and then shown a destination (New Goal Events) and asked to indicate direction to the goal. They then viewed footage in which their viewpoint traversed the street (Travel Periods) until arriving near the junction. Before entering new street segments (Street Entry) the participants pressed a button to indicate which direction at the upcoming junction provided the shortest path to the goal (Decision Points), after which the movie continued along the route. Routes were predetermined such that they generally followed the optimal route but occasionally required a forced detour where the movie travelled along a suboptimal path.

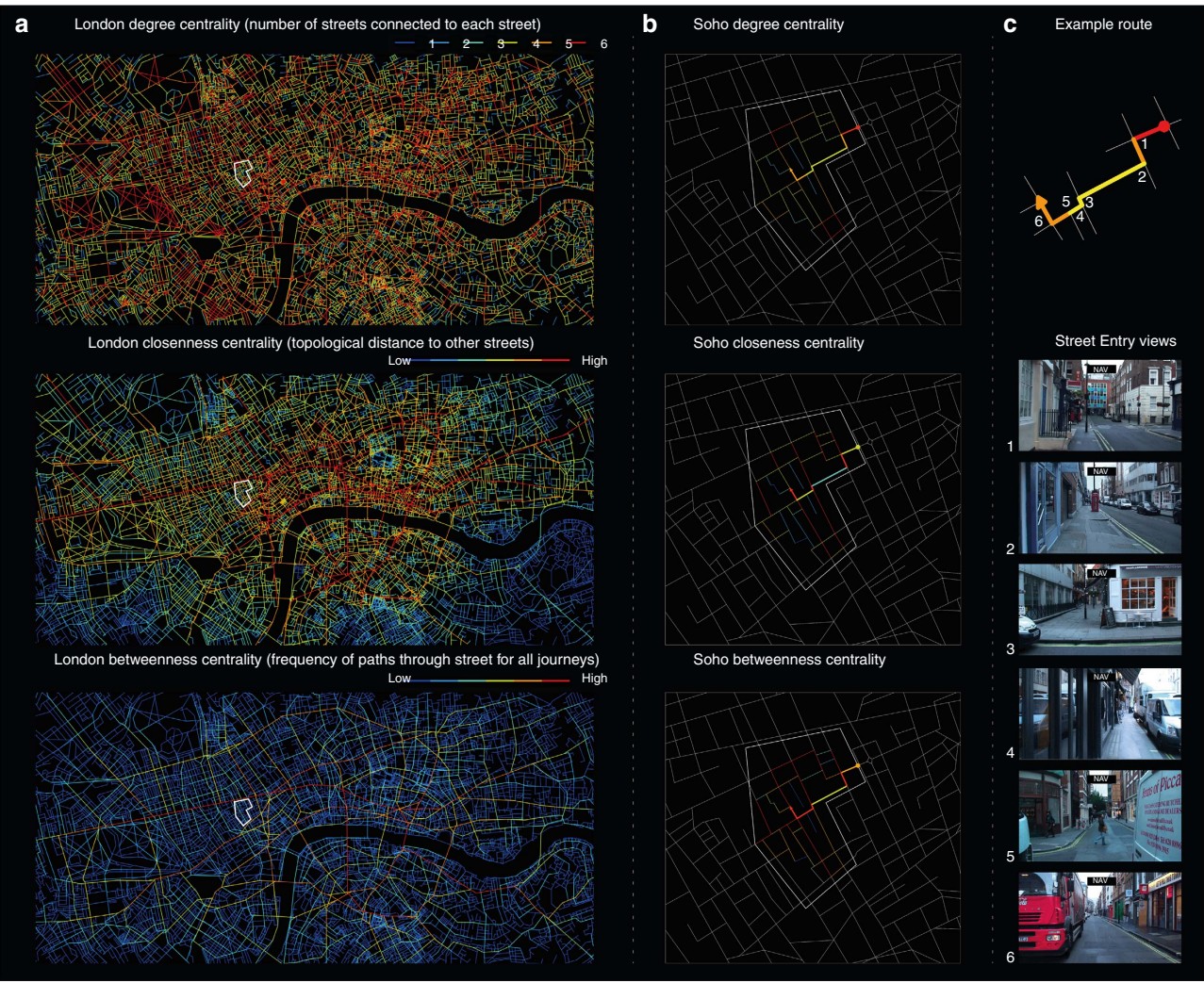

**Figure 2 | Graph-theoretic analysis of London (UK) street network centrality and the fMRI navigation task.** (**a**) Plots of central London (UK) street segment centrality measures (degree, closeness and betweenness). We used a segment-based approach known as space syntax. Here degree centrality measures the number of connecting segments to any segment, closeness measures how far any two segments are and betweenness measures the number of shortest paths from all segments to all other segments that pass through that segment. See Supplementary Table 1 for the relationship between measures in Soho. White bounded region in each plot indicates the region of Soho learned and navigated during fMRI scanning. See Supplementary Fig. 1 for the frequency of each value of centrality for Central London and this region of Soho. (**b**) Plots of segment centrality measures for the streets navigated in Soho. Thicker lines display an example of one of the 10 routes navigated during fMRI. (**c**) Top: degree centrality of the street segments in the example route plotted with each of the six Street Entry Events marked. Bottom: movie frames from our fMRI navigation task at the six Street Entry Events in the example route above.

Control routes had the identical format to Navigation routes, except participants were instructed not to navigate and told which button to press at Decision Points. Route and task were counterbalanced. Participants were 84.82% (s.d. = 10.96) correct at New Goal Events and 79.91% (s.d. = 13.28) correct at Decision Points[18].

We explored the fMRI data with three graph-theoretic centrality measures of the street segments: degree, closeness and betweenness. For an explanation of the measures see Fig. 1, Supplementary Fig. 1 and Supplementary Table 1. In previous research we have found hippocampal activity correlated with both raw spatial metrics (for example, distance to the goal) and the change in metrics (for example, the change in distance to the goal)[18]. Thus, we tested whether the hippocampal-processing demands might reflect the future simulation demands purely at Street Entry (raw values) or the change in demands that occurs at Street Entry (change in values).

**Posterior hippocampus tracks change in degree centrality.** Consistent with our hypothesis, we found that right posterior hippocampal activity was significantly positively correlated ($n = 24$, general linear model (GLM) $P < 0.05$ family-wise error (FWE)-corrected for region of interest (ROI)) with the change in the number of possible local paths (degree centrality) at Street Entry Events during Navigation routes (Fig. 3a,b). A significant posterior hippocampal response was observed whether the change in degree centrality was entered into our analysis as single parameter ($n = 24$, GLM $P < 0.05$ FWE-corrected for ROI; Fig. 3b), or when the changes in all three centrality measures were entered into an analysis ($n = 24$, GLM $P < 0.05$ FWE-corrected for ROI; Supplementary Fig. 2). We did not observe a similar response in the posterior hippocampus to the changes in betweenness or closeness centrality (Supplementary Tables 2 and 3), or a response to any raw centrality measure (Supplementary Table 4). A significant

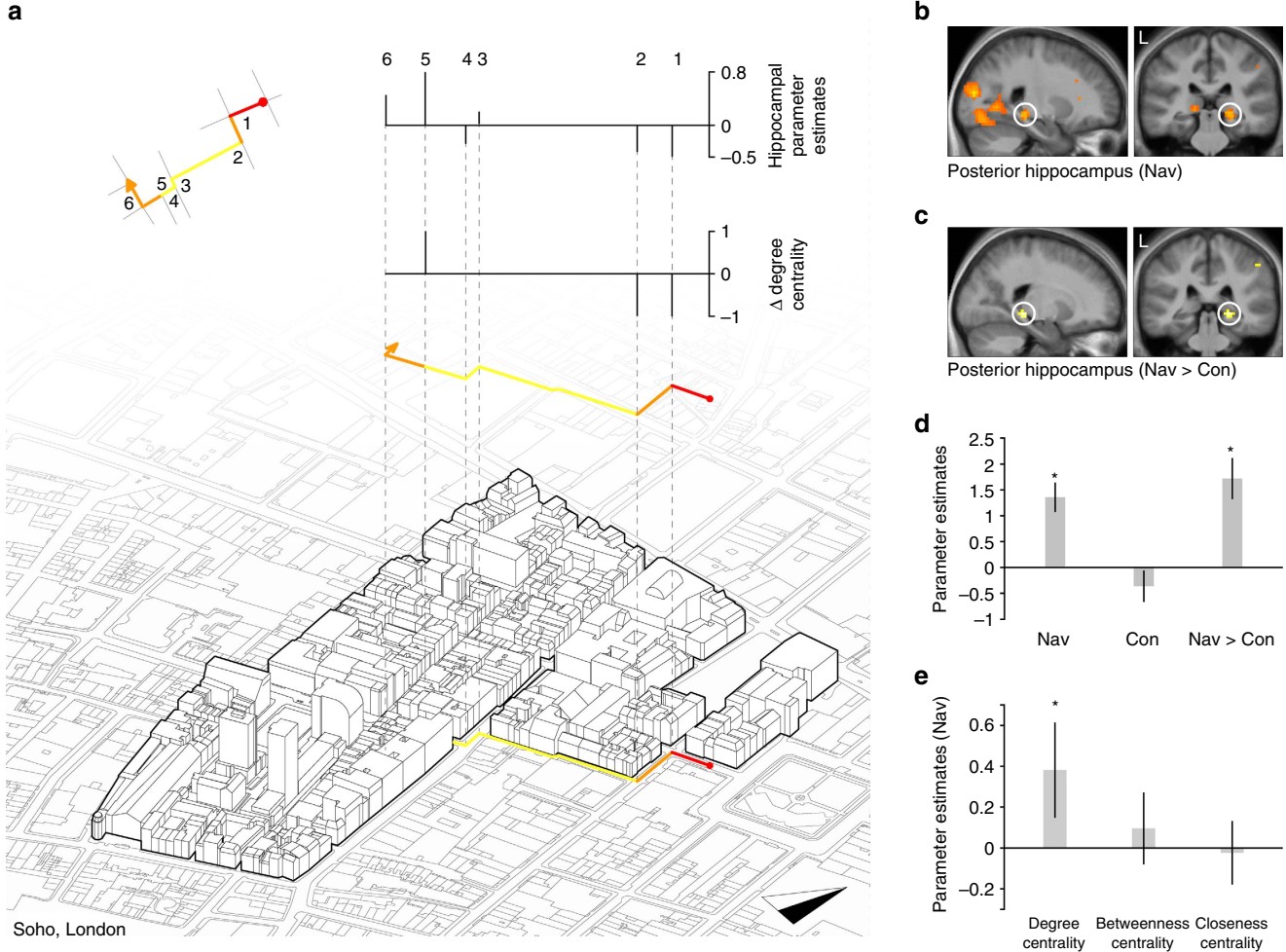

**Figure 3 | Posterior hippocampal activity is correlated with the change in degree centrality during navigation.** (**a**) Top left: degree centrality plotted for each street segment for an example route (see Fig. 2c). Right: axonometric projection of the buildings in Soho plotted on a map of Soho. Degree centrality of the route is plotted on the map and projected above. Above the route the graph plots the change in degree centrality for each boundary transition and the top graph plots the evoked response in the right posterior hippocampus at each of the individual boundary transitions (1–6). Analysis of this plot was not used for statistical inference (which was carried out within the statistical parametric mapping framework), but is shown to illustrate the analytic approach. (**b,c**) Right posterior hippocampal activity correlated significantly with the change in degree centrality for Navigation and Navigation > Control during Street Entry Events. Statistical parametric maps are displayed with threshold $P < 0.005$ uncorrected on the mean structural image. (**d**) Parameter estimates for the mean activity in the right posterior hippocampus ROI for Navigation ($t_{23} = 4.24$, $P = 0.0003$), Control ($t_{23} = 1.17$, $P = 0.25$) and Navigation > Control ($t_{23} = 4.64$, $P = 0.0001$) comparisons for a model containing categorical change in degree centrality (see Supplementary Table 2). (**e**) Parameter estimates for the mean activity in the right posterior hippocampus ROI for Navigation > Control condition for a model containing degree centrality ($t_{23} = 2.28$, $P = 0.03$), betweenness centrality ($t_{23} = 0.53$, $P = 0.59$) and closeness centrality ($t_{23} = 0.14$, $P = 0.88$) measures (Supplementary Table 3 and Supplementary Fig. 3). Error bars denote the s.e.m. See Supplementary Fig. 4C for anterior hippocampal ROI mean responses.

correlation only occurred when navigation was required; no significant correlation was observed during the Control routes ($n = 24$, GLM $P < 0.05$ FWE-corrected for ROI; parameter estimates for the mean activity in the right posterior hippocampus ROI for Navigation ($t_{23} = 4.24$, $P = 0.0003$) and Control ($t_{23} = 1.17$, $P = 0.25$); Fig. 3b,d). Furthermore, hippocampal activity was significantly more positively correlated with the change in degree centrality during Navigation routes than Control routes ($n = 24$, GLM $P < 0.05$ FWE-corrected for ROI; parameter estimates for the mean activity in the right posterior hippocampus ROI for Navigation > Control ($t_{23} = 4.64$, $P = 0.0001$; Fig. 3c,d). We also found that the posterior hippocampus was significantly more correlated with the change in degree centrality than the change in closeness centrality and, at a lower threshold, more correlated with the change in degree centrality than the change in betweenness centrality

(parameter estimates for the mean activity in the right posterior hippocampus ROI for Navigation > Control condition for a model containing degree centrality ($t_{23} = 2.28$, $P = 0.03$), betweenness centrality ($t_{23} = 0.53$, $P = 0.59$) and closeness centrality ($t_{23} = 0.14$, $P = 0.88$) measures; Fig. 3e and Supplementary Fig. 3). Thus, the right posterior hippocampus appears to track changes in local path options (degree centrality) when new streets are entered and only when navigating.

**Anterior hippocampus tracks change in closeness.** We found that activity in the right anterior hippocampus was significantly correlated with the change in closeness centrality at Street Entry Events during Navigation routes, but not during Control routes, and was significantly more correlated with the change in closeness during Navigation routes than Control routes

($n = 24$, GLM $P < 0.05$ FWE-corrected for ROI; Supplementary Fig. 4 and Supplementary Table 3). This was the case only when the changes in all three centrality measures were entered into the analysis. We did not find evidence that the anterior hippocampus was more correlated with the change in closeness centrality than the change in degree centrality or the change in betweenness centrality (Supplementary Table 3).

We considered that the hippocampal responses to the changes in centrality measures might be driven by visual properties of the environment rather than purely by centrality measures. Thus, we measured various visual properties of the environment that have been examined in prior studies examining graph-theoretic measurements of urban networks[19]: line of sight, street width, topological distance to edge of Soho, number of visible connecting streets, visible junctions and presence of shops, people or vehicles (Methods and Supplementary Tables 5–8). We found that, while none of our measures were significantly correlated with the change in degree centrality, the line of sight and the step depth to the boundary of Soho were correlated with the change in closeness centrality (nonparametric Spearman's correlation false discover rate-corrected $n = 24$, step depth to boundary $r = 0.37$, $P = 0.004$, and line of sight $r = 0.60$, $P < 0.001$; Supplementary Table 6). Thus, we examined whether the anterior hippocampal response was selective to the change in closeness centrality or driven by these other factors. We found that anterior hippocampal activity was not significantly correlated with the change in closeness when step depths to boundary or line of sight were included in the analysis ($n = 24$, GLM $P < 0.05$ FWE-corrected for ROI; Supplementary Fig. 4), nor was the anterior hippocampus significantly correlated with line of sight or step depth to boundary ($n = 24$, GLM $P < 0.05$ FWE-corrected for ROI). Thus, the anterior hippocampal response at Street Entry Events appears to reflect a combination of environmental properties that relate to the more global importance of a street, for example, streets closer to the centre of the network and which have a long line of sight.

**Specificity of the posterior hippocampal response**. While the change in degree centrality was not correlated with our measures of the visual properties of the environment, we nonetheless examined whether activity in the posterior hippocampus was selectively correlated with the change in degree centrality when accounting for the other measures of the visual properties of the environment. When these variables were entered into analyses with the change in degree centrality, we found evidence of a significant response to the change in degree centrality in right posterior hippocampal activity across the models ($n = 24$, GLM $P < 0.005$ uncorrected; Supplementary Fig. 5). Thus, the right posterior hippocampal response to the change in degree centrality is not explained either by other properties of the environment and appears to track the change in degree centrality.

A previous analysis of this data set[18] revealed that at Detours the change in the path distance to the goal was significantly correlated with activity in a slightly more posterior portion of the right hippocampus. Thus, we examined whether our observed hippocampal response at Street Entry Events was independent of changes in the path distance to the goal. We found no significant correlation between change in the path distance and the change in degree centrality (nonparametric Spearman's correlation $n = 24$, $r = 0.078$, $P = 0.569$). When both parameters were entered into an fMRI analysis, we found that hippocampal activity remained significantly correlated with the change in the degree centrality during Navigation routes

($n = 24$, GLM $P < 0.005$ uncorrected; Supplementary Fig. 6 and Supplementary Table 9). Thus, the right posterior hippocampal response to the change degree centrality is not simply explained either by changes in distance to the goal or by visual properties of the environment.

**Posterior hippocampal response is driven by retrieval**. Because it was difficult to observe all possible paths connected to a street segment at Street Entry Events (see examples in Fig. 2c), and the posterior hippocampal response to degree centrality was absent in Control routes, it seems likely that the hippocampal response was associated with retrieval of the network topology, rather than in response to visual properties of the stimuli. Nonetheless, we tested whether new participants (naive to Soho or trained experts) were able to detect changes in the degree centrality at each Street Entry Event purely by viewing our film simulation (see Methods). We found that naive participants could not reliably detect changes in degree centrality, whereas trained experts could (binomial test comparing the performance of the participants with chance level of 33.33% based on three possible choices at each decision point; $P = 0.243$ for naive participants and $P < 0.001$ for trained participants; Supplementary Table 10 and Supplementary Fig. 7).

**Street entry drives posterior hippocampal response**. We found that the right posterior hippocampal response was specific to Street Entry Events ($n = 24$, GLM $P < 0.05$ FWE-corrected for ROI; Fig. 4). No significant correlations between the change in degree centrality and hippocampal activity were observed during events sampled in Travel Periods or Decision Points ($n = 24$, GLM $P < 0.05$ FWE-corrected for ROI). Moreover, the right posterior hippocampal activity was significantly more correlated with the change in degree centrality during Street Entry than during these other events (comparison of right posterior hippocampal activity at Street Entry Events and at Decision Points ($t_{23} = 2.34$, $P = 0.02$) or at Travel Period Events ($t_{23} = 4.01$, $P = 0.001$); Fig. 4). Hippocampal activity was not significantly correlated with the change in degree centrality at Decision Points whether the change was calculated with respect to the previous street segment or the future segment ($n = 24$, GLM $P < 0.05$ FWE-corrected for ROI).

**Prefrontal activity reflects planning demands at Detours**. To explore whether prefrontal activity was specifically related to planning future paths, we examined whether responses were correlated with measures in a BFS-planning approach[13]. In these models the planning demands are calculated from the sum of the degree centrality in the future street segments to be travelled through to reach the goal (first level of the search tree), such that the more possible paths in the future streets, the greater the demands on planning (Fig. 5a and Methods). In our post-scan debriefing (see Methods) we found that participants reported more planning at Detours than at Decision Points, often reporting that they had planned their choice before the Decision Point. Consistent with this pattern, and with our theoretical prediction that lateral PFC regions might be responsible, we found that bilateral inferior lateral PFC was significantly correlated with our measure of BFS-planning demands in the first layer of the street network at Detours, but not at Decision Points, and significantly more correlated with planning demands at Detours than Decision Points ($n = 24$, GLM $P < 0.001$ uncorrected for ROI; Fig. 5b and Supplementary Table 11). We also found that planning demands did not significantly correlate with prefrontal activity during detours in Control routes, where participants were instructed to select one path and the route continued along a different path

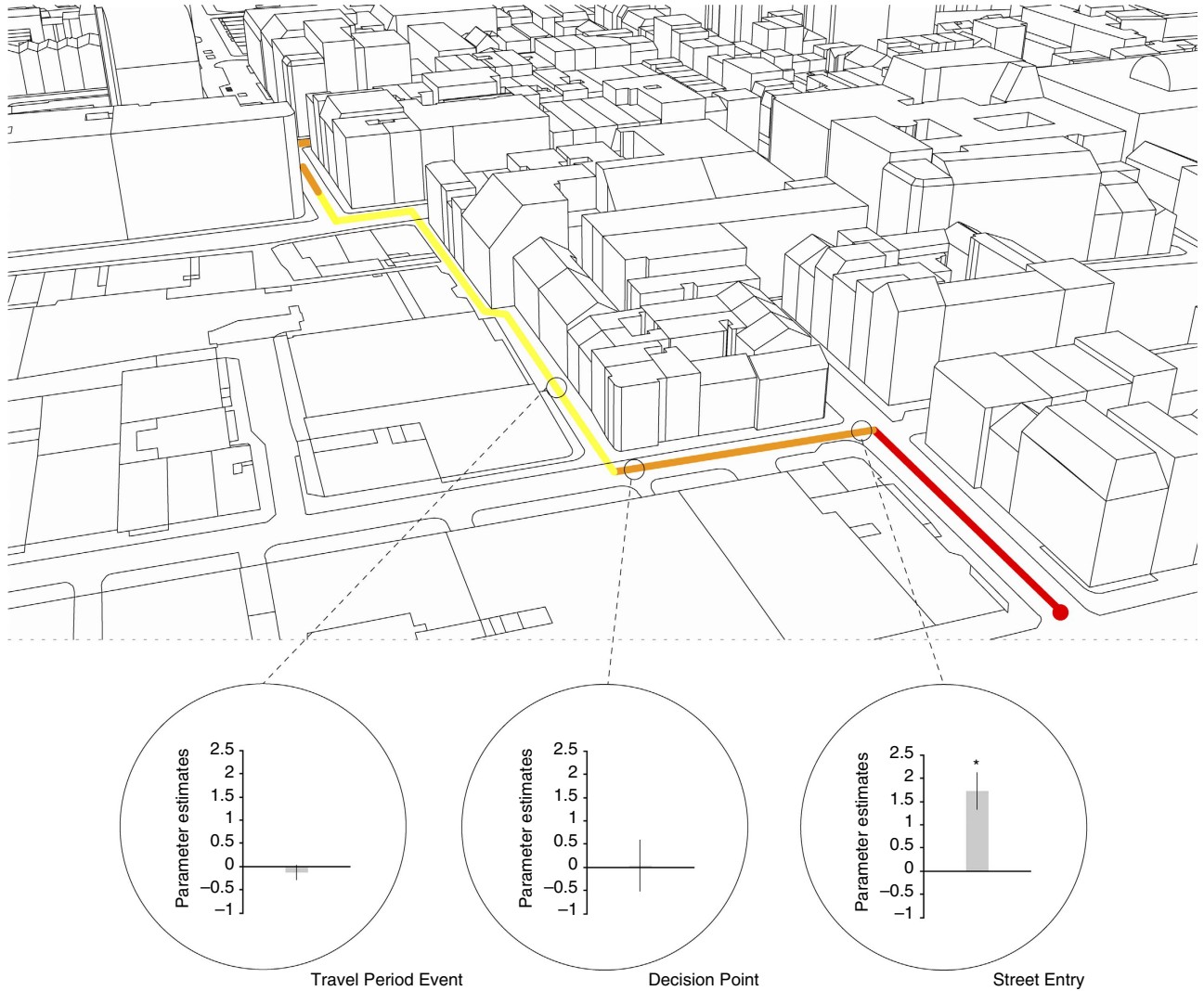

**Figure 4 | Posterior hippocampal activity correlated with the change in degree centrality specifically at Street Entry Events.** Top: perspective view of Soho showing part of the example route (Fig. 2a) shown to illustrate the three examples of the different time points examined. During navigation routes, right posterior hippocampal activity was significantly more correlated with the change in degree centrality at Street Entry Events than at Decision Points ($t_{23} = 2.34$, $P = 0.02$) or at Travel Period Events ($t_{23} = 4.01$, $P = 0.001$), *Significance at a threshold of $P < 0.05$ corrected for ROI. Error bars denote the s.e.m.

($n = 24$, GLM $P < 0.001$ uncorrected for ROI). The prefrontal response was significantly more correlated with planning demands in Navigation route Detours than Control route Detours ($n = 24$, GLM $P < 0.001$ uncorrected for ROI; Supplementary Table 11). We found that no significant activity correlated with the planning demands when the first layer (Fig. 5a) and the second layer of the network were combined to calculate planning demands, indicating that lateral PFC activity reflects the number of path choices in the street segments immediately beyond the next junction, rather than an extensive search of all streets two choices ahead in the network ($n = 24$, GLM $P < 0.001$ uncorrected for ROI). We also found that no significant activity in the hippocampus correlated with the BFS-planning demands whether calculated at the first level of the search or both first and second levels of search ($n = 24$, GLM $P < 0.001$ uncorrected for ROI).

Outside our frontal and hippocampal ROIs we found no regions significant when correcting for whole-brain volume. For completeness we report all regions active in contrasts at an uncorrected threshold of $P < 0.001$ in Supplementary Tables.

## Discussion

In summary, we show evidence that when entering a street during navigation the right posterior hippocampal activity tracks changes in the number of available path options (degree centrality), the right anterior hippocampus tracks changes related to the closeness centrality of the street and, at forced detours, lateral prefrontal activity scales with the planning demands consistent with a BFS of the street network. These discoveries will help shape models of how the hippocampus and PFC support navigation, memory and future simulation, which have hitherto generally neglected the importance of entering new regions of space and the processing of topology.

Our observation that posterior hippocampal activity was correlated with the change in degree centrality is consistent with the idea that the hippocampus re-activates representations of paths[18,20], with the more paths requiring re-activation the more activity elicited in the hippocampus. Such processing of the local streets is in agreement with the view that the hippocampus helps simulate future possible options to guide choices[1–5]. Hippocampal 'replay' or 'forward-sweeps'[8,21–23] in the dorsal

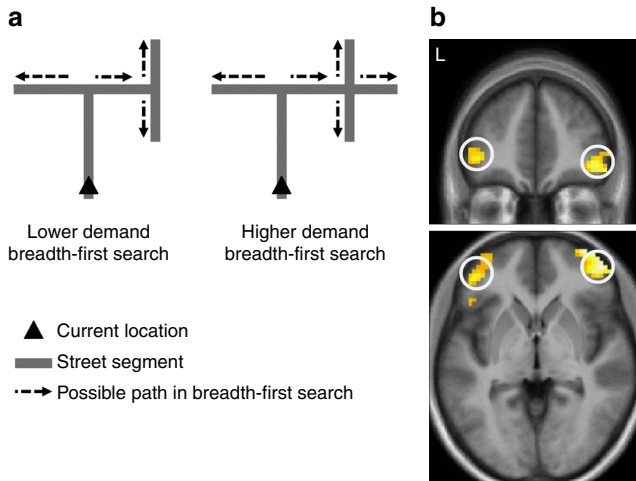

**Figure 5 | Inferior lateral prefrontal activity correlates with the demands of a breadth-first search at Detours.** (**a**) Diagrams of an example street network contrasting scenarios of lower and higher demand breadth-first search. Breadth-first search assumes the search space (street segments) as a tree and considers all possible solutions within one level before proceeding to the subsequent level. In these diagrams, covering the first layer of the search, the lower demand scenario shows less possible paths, while the higher demand scenario shows a greater number of possible paths. For details see Methods. (**b**) The statistical parametric map showing correlation ($P < 0.05$ FWE-corrected) of the left and right lateral PFC with planning demands for the first layer of the decision tree (Navigation>Control). We found bilateral lateral PFC activity correlated with planning demands ($P < 0.001$ uncorrected) during Detours in navigation routes, but not in control routes. We found no significant correlations when the planning demands of first and second layer combined were entered in the analysis. The statistical parametric maps are displayed on the mean structural image at a threshold of $P < 0.005$ uncorrected and five voxels minimum cluster size. See Supplementary Table 11 for details of activations. Comparison of parameter estimates of peak voxel in the right lateral PFC showed a significantly greater response at Detours compared with Decision Points ($t_{23} = 3.49$, $P = 0.002$).

hippocampus (homologue of the posterior hippocampus) may be the mechanism by which the paths are re-activated; indeed, hippocampal replay has been shown to reflect the topology of the environment[8]. The pattern of our data helps clarify between two possible conceptual models of how the hippocampus might process future paths during navigation. Because we found that hippocampal activity reflected the change in degree centrality, not the raw degree centrality, a model in which the hippocampus only processes future paths the moment a street is entered is not consistent with our data. Rather, our data agree with a model in which the hippocampus simulates possible paths throughout the journey, with the hippocampal activity we observed reflecting the increase or decrease in the number of potential future paths to be re-activated as each new street is entered.

A right hippocampal locus dovetails with prior work highlighting the importance of the right hemisphere in spatial processing[18,24–28]. Given recent evidence of social space coding in left hippocampus[29], an intriguing possibility is that the left hippocampus may play an equivalent role in processing the complex topologies of the social networks humans are required to navigate in daily life.

Our data provide some fuel for the debate on the functional differentiation of long axis of the hippocampus[25,26]. Observation of a posterior hippocampal response to local spatial properties

(degree centrality) and an anterior hippocampal response to more global spatial properties (closeness centrality, see Fig. 1) is consistent with the proposal that the posterior region codes fine-grain detail and the anterior codes global information[18,30–32]. While the posterior hippocampal response is consistent with a local replay of the path representations, an anterior hippocampal response to the change in closeness centrality may reflect a different mechanism. For example, the anterior hippocampus may integrate information during learning about the transition structure across the street network to aid optimal navigation, for example, which streets will lead to the centre of the network. Consistent with this, recent evidence indicates that the anterior hippocampus may represent the graph community structure during learning the nature of transitions between a set of arbitrary stimuli[33]. Notably, our analysis revealed that the anterior hippocampal response was not selective to closeness centrality, and may represent information more generally about the important streets in the environment, with 'important' defined here by how central it is in the network both in relation to all streets (closeness), the edge of the space (step depth to boundary) and what can be seen from that street (line of sight). Our results also have implications for city planning by showing, in line with previous studies[16,19,34], that certain visual properties of the environment, especially how far one can see (line of sight), but also street width and presence of people, are related to centrality in the city. Future fMRI research with tailored virtual environments will be useful to understand what properties of the environment drive activity in the anterior hippocampus. Past research, for example, indicates that the contextual uncertainty of the environment may be important in eliciting anterior hippocampal responses[35,36].

A recent model exploring how state space should be optimally segmented for planning has revealed that degree centrality measures provide better optimality for planning, rather than segmentation of the state space by closeness or betweenness[37]. Thus, our finding of posterior hippocampal representations of the change in degree centrality, rather than betweenness or closeness, may relate to optimal retrieval of information for planning. While the hippocampus appears to represent information about changes in topological properties, the lateral prefrontal activity reflected the demands of searching the network of possible future paths when re-planning was required at Detours. This is consistent with prefrontal regions playing a role in spatial planning during navigation[10,38,39]. However, it has not been clear which regions of PFC are central to this function. We have previously argued that lateral frontopolar regions may be important[10]. This proposal was based in part on the observation of increased lateral frontopolar activation in London taxi drivers during re-planning at forced detours when navigating a virtual simulation of London[40]. Here we show that activity in this same region, rather than simply being active at Detours, is correlated with the path-planning demands. Given that the PFC is thought to be domain general in its processing[9], it seems likely that the lateral PFC regions we have identified here would be engaged during other tasks that require searching a decision tree.

Here we examined how brain regions support navigation by processing topological properties of a recently learned street network that lacked hierarchical structure. It is possible that prolonged exposure to the environment would drive an increase in global processing of network topology in the hippocampus, or a switch to topological processing in cortical regions. In light of recent discoveries of the brain regions that support navigation of subway networks[38], it is possible that learning to exploit the hierarchical structure leads to the transfer of planning from a BFS in lateral PFC to a more efficient hierarchically organized plan mediated by dorsomedial PFC and premotor cortex.

## Methods

**Participants.** Twenty-four right-handed, healthy participants (11 female, mean age: 26.25 years, s.d.: 3.52, range: 20–35 years) with no history of neurological disease and normal or corrected to normal vision took part in this experiment. Participants gave written informed consent, and the study was approved by the University College London (UCL) ethics committee.

Eligibility for the experiment was assessed across all participants using two screening criteria: existing knowledge of the Soho testing environment and navigational ability. Only participants who reported minimal or no experience with the environment were invited to take part in the study. Participants were required to score above 3.6 using the Santa Barbara Sense of Direction Scale (one s.d. below the mean score provided by Hegarty et al.[41]). The mean Santa Barbara Sense of Direction score across participants was 4.89 (s.d. 0.69).

**Stimuli and apparatus.** An area of London (Soho) including 26 streets was selected as the testing area. This specific area was selected because of its high density of streets and large number of distinct locations such as pubs and shops. Twenty-three goal locations were specified. For details of the area, map and goal locations refer to Howard et al.[18]. Ten testing and three training routes were defined and filmed (using a HD Sony Z1 and a B Hague camera stabilizer). Videos were edited to create two sets of movie stimuli (Navigation and Control) used in the experiment. During the Navigation videos, onscreen instructions asked participants to actively think about routes, while during the Control videos, onscreen instructions asked participants to press corresponding keys.

Stimuli were presented using MATLAB (v7.5, MathWorks) and the Cogent2000 toolbox (v1.28, www.vislab.ucl.ac.uk/cogent_2000.php). Responses were recorded using a response box positioned under the participant's right hand.

**Procedure.** The experiment consisted of studying a training pack, a training session and a testing session. One week before the training session participants were given a training pack to familiarize themselves with the layout of the test area, each streets' name, and also location and name of goal locations.

The training session happened 1 day before the testing session. During this session, participants were taken on a 2-h tour of the test area in Soho. During this tour their spatial knowledge was rigorously tested and feedback was given to maximize participants' knowledge of the area. The training route was designed so that (1) it was different with all the filmed routes, (2) each start location (locations at the beginning of the video footages) was visited once and (3) each goal location was passed at least twice and from different directions. When each of these locations was reached the experimenter showed participants the coloured photograph of the start or goal as well as their current position on a map. These coloured photographs were used in the video footage to indicate different locations. Immediately after the tour, participants were tested to assess their knowledge. Immediate feedback was provided to guide participants towards any aspects they should 'revise' on the final evening before scanning (for further details refer to Howard et al.[18]).

The testing session began with a brief training to ensure that participants understood the task requirements. A total of three training routes were viewed (two Navigation and one Control). Each video began with 12 s of fixation cross, followed by a 5 s of a cue word ('NAVIGATION' or 'CONTROL'), indicating the type of the following route. The words 'NAV' or 'CON' were presented on top of the screen throughout the presentation of the video. The video continued with presentation of a start image with a temporal jitter of 5–13 s. This start image indicated the current location and heading direction. The paths taken in the routes did not match the paths walked during training. Thus, to solve the navigation task participants could not simply recall a previously walked route or sequence of actions, but rather had to construct novel sequences through the space.

Two events were included in the videos during which the video was paused: New Goal Events (NGE) and Decision Points (DP) with 9 and 5 s, respectively. A colour photograph of the new goal was presented during NGE. This presentation contained an initial 4 s with a text describing its location. This followed with 5 s asking participants to indicate the location ('goal L/R?') with regard to the current heading direction in Navigation condition and asking participants to indicate whether one can buy drink from that location in Control condition. DP occurred a few seconds before each junction. In the Navigation condition participants were presented with the option to turn at the junction ahead or go straight (for example, turn L/R?), while in the Control condition they were asked to press the button corresponding to the optimal path (for example, press left button). The amount of time between DP and the onset of the following turn or junction crossing (Street Entry Events) were temporally jittered to last between 3 and 9 s to allow separate measures of the BOLD signal at these two events. After each turn at the beginning of each new street section text appeared onscreen for 3 s describing the current location and general cardinal heading direction (for example, Broadwick St facing east). For some of the Street Entry Events (46.15 or 51.85% depending on the combination of routes), the route was suboptimal for reaching the current goal and participants were thus forced to take a detour to the goal (Detours). The mean duration of the routes was 266.60 s (s.d. = 43.63, range = 198–325). Routes were presented at walking speed (mean = 1.6 m s$^{-1}$, s.d. = 0.41). Ten routes and task (Navigation/Control) were counterbalanced across participants. For further details refer to ref. 18.

Immediately post scan, participants took part in debriefing session outside the scanner in a testing room. Participants were not warned in advance that this would occur. In this debriefing session participants re-watched the five Navigation routes they had experienced on a laptop (12 inch screen) in a similar manner to ref. 40. At each of the events (NGE, DP, Street Entry) the film was paused and participants were asked to describe whether they remembered planning or thinking about their future route.

**Graph theory analysis.** A set of formal analytic measures of the environmental layout, based on graph-theoretic measures used in the field of space syntax, were used. These measures examine different properties of centrality in the street network. Space syntax methods relate human behaviour to the layout of the environment[15,16]. These methods provide a formal way of analysing the spatial properties of an environment, and can be applied to both indoor and outdoor spaces. A number of different methods fall under the term 'space syntax', and can be applied at different scales (for example, local/global). For space syntax analyses relating to the street network, the street network is represented as a graph. Graph-theoretic approaches have been adopted by a number of built environment disciplines as a way of analysing the relationship between spaces[42]. There are two ways of translating information in the built environment into a graph, resulting in primal or dual graphs. The appropriate type of graph must be matched with the type of analysis. Primal graphs are concerned with information at street intersections: street junctions are the nodes in the graph, and streets as the links between the nodes. This results in a graph that closely matches the geographic urban layout. Dual graphs focus on the streets themselves (as opposed to street junctions). This type of graph is relevant for street network analysis: street segments are the nodes in the graph, and the connections between street segments are the links between the nodes. Dual graphs highlight the topological properties of the network and tend not to resemble the map of the physical location. Space syntax analysis is based on a dual graph representation of the street network, also known as a dual network. A number of different graph-theoretic measures can be applied to such a graph to examine properties of centrality. Typically, three graph-theoretic measures of centrality are used: degree centrality, closeness centrality and betweenness centrality. Figure 1 provides an illustration of how these three measures capture different properties of an example street network. In the below, 'segments' refer to the units of street sections that form the dual graph.

Degree centrality measures the total number of edges connected to any node. Applied to the urban network, degree centrality is the number of connecting street segments to any street segment.

Closeness centrality is defined in ref. 17 as:

$$C_C(p_i) = \left( \sum_k d_{ik} \right)^{-1}$$

where $d_{ik}$ is the length of a geodesic (shortest path) between node $p_i$ and $p_k$. Applied to an urban grid, closeness centrality is the reciprocal of the sum of the topological distance from that segment to all other segments. It reflects how likely it is that a segment is an origin or destination segment.

Betweenness centrality is the number of shortest paths from all segments to all other segments that pass through that segment. It is based on the measure defined in ref. 43:

$$C_B(p_i) = \frac{\sum_j \sum_k g_{jk}(p_i)}{g_{jk}} (j<k)$$

where $g_{jk}(p_i)$ is the number of geodesics between node $p_j$ and $p_k$ which contain node $p_i$ and $g_{jk}$ the number of all geodesics between $p_j$ and $p_k$. It reflects the likelihood that a segment is an intervening space in between an origin and a destination.

Space syntax analyses, based on graph-theoretic measures of the street network, have linked pedestrian movement to the topological properties of the street network. This has been shown for both for aggregate pedestrian movement[44,45] and for the navigational decisions made by individuals[19]. The methodology is robust when compared to observed pedestrian flows across locations, scales, cities and cultures[16]. The approach is based solely on an analysis of the topological properties of the street network; no other information is included in the model. It has been suggested that part of the reason why space syntax analyses are so successful is that these types of analyses pick up on elements that are naturally processed during cognition[46]. It would seem that people intuit how connected a particular street is within the street network as a whole[46,47].

On the basis of past research we considered that properties of the environment[14–17,20] or the distance to the goal[2] might correlate with our centrality measures or the change in centrality. Such factors might in themselves drive hippocampal activity at Street Entry Events. Thus, we measured a number of properties of the streets and the distance to the goal, and examined them in relation to our fMRI analysis—these measures are outlined below.

The following measures are recorded directly from the first-person videos. They reflect what can actually be seen from a certain point in the videos, as opposed to what could theoretically be the case. Obstacles and obstructions present in the videos are taken into account, so that the parameters reflect the information available to participants at a given point in the video.

**Table 1 | Events/epochs of interest and their duration.**

| No. | Effect | Duration |
|---|---|---|
| 1 | Task epochs | 198–325 s* |
| 2 | Street Entry | 0 |
| 3 | New Goal Event | 9 s |
| 4 | Decision Point | 5 s |
| 5 | Travel Period Events | 0 |

These were included separately for navigation and control routes and were included in all the models. Travel Period Events were time points during the travel periods equidistant between the other events.
*Varied across routes.

**Table 2 | General linear models reported in this article.**

| Model | Time period | Modulatory parameters | Table | Figure |
|---|---|---|---|---|
| 1 | Street Entry | degree centrality | S4 | |
| 2 | Street Entry | [Δdegree centrality] | S2 | 3 |
| 3 | Street Entry | betweenness centrality | S4 | |
| 4 | Street Entry | closeness centrality | S4 | |
| 5 | Street Entry | [Δbetweenness centrality] | S2 | |
| 6 | Street Entry | [Δcloseness centrality] | S2 | |
| 7 | Street Entry | [Δdegree centrality] | S3 | S2–S4 |
| | | [Δbetweenness centrality] | | |
| | | [Δcloseness centrality] | | |
| 8 | Street Entry | [Δdegree centrality] | | S5 |
| | | [ΔPOI]* | | |
| 9 | Travel Period Events | [Δdegree centrality] | | 4 |
| 10 | Decision Points | [Δdegree centrality] | | 4 |
| 11 | Street Entry | [Δdegree centrality] | S9 | S6 |
| | | [Δpath distance at detours] | | |
| 12 | Street Entry | [Δdegree centrality]† | | S7 |
| 13 | Street Entry | BFS for degree centrality | | 5 |
| 14 | Street Entry | BFS for betweenness centrality | | 5 |
| 15 | Street Entry | BFS for closeness centrality | | 5 |

BFS, breadth-first search.
General linear models indicate the time point of the event (time period, see Table 1), the modulatory parameters and their reference to tables and figures in the main manuscript and supplementary documents.
Models 1 and 2 were conducted to examine our main question of interest. Subsequent models were control analyses conducted to determine the specificity. Δparam refers to *change* of value between previous segment and current segment (value at current segment minus value at previous segment). [Δparam] refers to categorical change of param with −1 for Δparam < 0, 0 for Δparam = 0 and 1 for Δparam > 0.
*POI refers to other parameters of interest: visible junction, visible connecting street, path distance, Euclidean distance to goal, step depth to goal, step depth to boundary, light of sight, street width, street length, number of visible people, number of visible vehicles and number of visible shops.
†For this model events in which [Δparam] = 0 was excluded. This was conducted as a follow-up to our behavioural experiment, see Methods.

*Number of visible connecting streets*. This is the actual number of visible path options from a given location. In contrast to the degree centrality measure, which records the number of connecting streets irrespective of whether they can be seen or not, this measure records what can actually be seen. It is similar to the visible connectivity measure used in Emo[48].

*Number of visible junctions*. This is the number of junctions visible from a given location. In contrast to number of visible connecting streets, this measure records the number of junctions in sight regardless of type of how many streets at each junction.

*Line of sight*. This is the longest line of sight measured in real-world meters from a given location. The line of sight, measured at eye height, is translated into a line on Ordnance Survey map of Soho. It is irrespective of the choice of route (if available). Many studies in the spatial cognition literature suggest that depth of view is critical for navigation[34,49,50].

*Street width*. This is the actual street width of the given location, measured in real-world meters. The location is translated onto the Ordnance Survey map of Soho.

*Presence of shops/people/vehicles*. This records the presence or absence of shops/people/vehicles from a given location. The presence of shops, people and vehicles are cues that convey how busy a street is. They are attractors in that a busy street is likely to have more of each. Research suggests that these elements are related to centrality measures of streets[44,51,52], and that people detect such cues during navigation[19,53].

*Step depth to Goal*. This is the optimal number of street segments required to reach the goal, starting from the current street and irrespective of the route taken in the video. For example, a destination on an adjacent street has a step depth of 1, as it is one street away. A topological step is counted at each junction, so that a destination lying exactly on the other side of a junction, but straight ahead, still has a step depth of 1.

*Step depth to Boundary*. Similar to the 'step depth to goal' parameter, this is the optimal number of street segments required to reach the boundary of the study, starting from the current street and irrespective of the route taken in the video.

For analyses examining the relation between these parameters see Supplementary Tables 5–8.

**fMRI acquisition and analysis.** Participants were scanned at the Birkbeck-UCL Centre for Neuroimaging (BUCNI) using a 1.5 Tesla Siemens Avanto MRI scanner (Siemens Medical Systems, Erlangen, Germany), with a 32-channel head coil. Functional scans were acquired using a gradient-echo echoplanar imaging sequence ( repetition time (TR) = 2.897 ms, echo time (TE) = 50 ms, flip angle = 90°, field of view (FoV) = 192 mm²). In each volume 34 oblique axial slices, approximately perpendicular to the hippocampus (64 × 64 × 34 matrix size) and 3 mm thick, were acquired (3 × 3 × 3 mm voxel size). Following this a high-resolution T1 structural scan was acquired (MPRAGE, 176 slices, 1 × 1 × 1 mm resolution). The first six functional volumes of each session (dummy scans) were discarded to permit T1 equilibrium. Statistical parametric mapping (SPM12, Wellcome Trust Centre for Neuroimaging, London, UK) was used for spatial preprocessing and subsequent analyses. Images were spatially realigned to the first volume of the first session to correct for motion artefacts, co-registered with the structural scan, normalized to a standard EPI template in Montreal Neurological Institute space and spatially smoothed with an isotropic 8 mm full-width at half-maximum Gaussian kernel filter. After preprocessing, the smoothed, normalized functional imaging data were entered into a voxel-wise subject-specific GLM (that is, the first-level design matrix). The regressors of interest and six subject-specific movement parameters (included as regressors of no interest) derived from the realignment phase of preprocessing were included in all the models. The effects of

interest are shown in Table 1. The periods of fixation between blocks were not modelled and treated as the implicit baseline. Each of the regressors of interest was then convolved with the canonical haemodynamic response function, and a high pass filter with a cutoff of 128 s was used to remove low-frequency drifts. Temporal autocorrelation was modelled using an AR(1) process. For effects with duration zero we took the standard approach of modelling events used in SPM12. This stick function is then convolved with the haemodynamic response function[54].

At the first level, linear-weighted contrasts were used to identify effects of interest, providing contrast images for group effects analysed at the second (random-effects) level. In a series of GLM analyses we probed the fMRI data with the parameters of interest and covariates, Table 2. Initially we examined degree centrality because it is the simplest topological measure and has been highlighted as important for path planning[37]. We examined the parametric modulation of degree centrality at Street Entry Events and the categorical change in this parameter ($\Delta$degree centrality) as the degree centrality (models 1 and 2). We examined the categorical change (1,0 or $-1$) in degree centrality because the range of variation in the change was highly limited. To establish that the observed response was unique to degree centrality, we probed the fMRI data with our measures of closeness centrality and betweenness centrality (models 3 and 4). To further separate the effects of these three parameters, we examined a model that included categorical value of change of all these parameters (model 7). Finally, to determine the specificity of the response in separate models we investigated correlation of [$\Delta$degree centrality] and one of the following covariate parameters of no interest [$\Delta$Visible Junction], [$\Delta$Visible Connecting Street], [$\Delta$path distance], [$\Delta$Euclidean distance to goal], [$\Delta$step depth to goal], [$\Delta$step depth to boundary], [$\Delta$light of sight], [$\Delta$street width], [$\Delta$street length], [$\Delta$presence of visible people], [$\Delta$presence of visible vehicles] and [$\Delta$presence of visible shops]. We tested whether it is possible to construct a single model including all the mentioned parameters, which was not possible, as the model could not be estimated in SPM. To investigate specificity of the correlation of [$\Delta$degree centrality] with activity of the right posterior hippocampus to Street Entry Events, we conducted an analysis where this parameter was also modelled at Travel Period Events and Decision Points. For Decision Points we examined both the change in centrality from the prior segment to the current segment the Decision Point was located in and the change that would occur after the outcome of the Decision Point (future segment—current segment).

Parametric regressors were not serially orthogonalized, thus allowing each regressor to account independently for the response at each voxel[55]. Each GLM explored the first-order parametric modulation of the events of that type, for both Navigation and Control routes. All models contained all the key events (see Table 1), plus Navigation and Control task blocks.

We focused our analysis on the right hemisphere because the right medial temporal lobe has been more consistently associated with spatial memory in humans (see, for example, refs 25,27,28,56–58). Thus, we created a ROI in the right hippocampus using the SPM Anatomy toolbox (Forschungszentrum Jülich GmbH). Statistical analyses of the mean responses in the ROI were conducted in SPSS. For SPM analysis we used the ROI for a small volume correction applying a threshold of $P < 0.05$ FWE. For follow-up analyses that involved a reduced numbers of events, such as when examining the subset of Street Entry Events that were Detours, we used a threshold of $P < 0.01$ uncorrected within the ROI. For completeness, we report all brain regions at a threshold of $P < 0.001$ uncorrected (or $P < 0.005$ for medial temporal lobe regions) and minimum of five contiguous voxels for the planned contrasts as we have done in prior work[18,59].

To further characterize the response post hoc we sectioned hippocampus into anterior and posterior ROIs. We used the MarsBaR SPM toolbox (v0.43, marsbar.sourceforge.net) to extract BOLD mean responses in the posterior and anterior hippocampal sections[60].

To analyse a measure of how search might occur as opposed to just detecting the future possibilities, we calculated the demands in a BFS in graph theory, which is a method for searching a graph[13]. We ran two levels of search with (1) sum of the degree centrality measures of all street segments connecting to the next immediate junction (see Fig. 5a) and (2) the combined sum of the degree centrality measures of all street segments connecting to the next immediate junction and the sum of degree centrality measures for all street segments connecting to the subsequent junctions on the optimal path to the goal. BFS assumes calculation based on the degree centrality; however, we considered whether the search demands might change if calculated with closeness or betweenness centrality. We found that BFS demand measures using degree centrality, closeness centrality or betweenness centrality were highly correlated ($r > 0.8$), and resulted in nearly identical SPM results to those from BFS using degree centrality. To test whether lateral PFC was involved in this search we created a lateral frontal ROI that encompassed the regions predicted in our recent review[10] using the bilateral inferior and mid-lateral frontal ROIs from the WFU_PickAtlas[61].

**Behavioural study.** Because the hippocampal response to the degree centrality was absent in Control routes, we reasoned that hippocampal response observed in Navigation routes might be related to retrieving information about the environment in order to aid navigation of future paths. If this is true then people who have knowledge of the streets should be able to determine whether degree centrality increases or decreases at Street Entry Events, and likewise those with no

prior knowledge should be unable to determine whether it has increased or decreased. To test this two experts with extensive knowledge of the environment from the training protocol, and a group of 11 naive participants (six male participants, age range 20–28 years) who reported minimal or no prior experience were tested on their ability to judge whether degree centrality increased, decreased or did not change at Street Entry Event in the fMRI study. Participants viewed the 10 routes tested in our fMRI task, and also two of the training routes to train them on our behavioural task. At each Street Entry Event participants were asked to press one of three buttons to make the judgement. Participants were told that 'a street segment is the part of a street between any junctions; for example, Oxford street in London is one long street, but is made up of many segments'. The two training routes were used to familiarize them with the idea and confirm that they understood the task. Participants' responses were recorded and marked based on the correct topological values to create a performance value for each participant. We estimated that because participants who were uncertain would be likely to opt for a 'no change' response, and because 'no change' was more often correct in the street network (59% of events), participants who were uncertain would potentially perform above chance, despite no knowledge. Thus, we examined the responses of participants for only those events during which degree centrality increased or decreased. Finally, to determine whether the right posterior hippocampal response was still correlated with the change in degree centrality in this subset of events we examined a GLM in which only events in which degree centrality increased or decreased were included.

**Data availability.** All the material will be available on request from the corresponding author.

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

## Acknowledgements

This work was supported by the Wellcome Trust (grant 094850/Z/10/Z to H.J.S.), James S. McDonnell Foundation (H.J.S.) and the Biological and Biotechnical Research Council (L.R.H.). We thank Dishad Husain and Jack Kelley for help with film production, Kleopatra Kouroupaki for help with initial analysis, Ravi Mill and Laura Morrison for task development, and Dharshan Kumaran, Kate Jeffery and Daniel McNamee for comments on the manuscript.

## Author contributions

H.J.S and L.R.H. conceived and designed the experiments. L.R.H., R.K. and F.E.Z. collected the data. A.-H.J., B.E., L.R.H., Y.Y. and J.P.S. analysed the data. H.J.S, A.-H.J and B.E. wrote the manuscript. All authors discussed the results and contributed to the manuscript.

## Additional information

**Competing financial interests:** The authors declare no competing financial interests.

