## [Peer Review File · Nature Communications]

Reviewers' comments:

Reviewer #1 (Remarks to the Author):

In this manuscript, the authors present evidence that the right posterior hippocampus codes for the topological connectedness of recently-entered path segments while navigating through an environment. Participants received extensive training in the Soho neighbourhood of London by taking a two-hour tour of the area and learning 23 goal locations. An analysis of Soho's street network provided graph-theoretic measures that were used to test whether brain regions were sensitive to topological properties of the environment. The next day, participants underwent functional MRI while viewing video clips of movement toward a goal along a route composed of street segments and junctions in Soho. The authors found that hippocampal activity increased when participants transitioned from a street segment with fewer connecting segments to one with more connecting segments; they also found that hippocampal activity decreased when transitioning from a segment with more to one with fewer connections. This effect was found to be robust when taking into account visual environmental properties, suggesting that participants retrieved environmental topology from memory when entering a new space.

I believe that this manuscript represents an important step forward in the way that researchers should approach investigation of the neural mechanisms underlying spatial memory and navigation in humans. Particularly, with this manuscript the authors demonstrate that in order to fully understand the neural mechanisms involved in human spatial navigation, it is crucial to take into account environmental properties in addition to individual differences and task demands. However, the Authors should address the issues highlighted hereafter before this manuscript can be considered for publication in Nature Communications.

Major issues:

- While I can appreciate that this manuscript presents novel and interesting findings, the Authors do not convincingly sell the findings in the discussion. It is unclear how these data are likely to change how we think about the role of the hippocampus in the context of navigation and spatial memory. The Authors use the title to suggest that the hippocampus is simulating the future when entering into new spaces, and they provide supporting references for this idea in the introduction, but fail to incorporate this idea into the discussion. I would suggest re-framing the concluding remarks to both generally highlight the novelty and importance of the findings, and specifically discuss the putative future simulation function that the hippocampus may be demonstrating in the experiment.

- I am skeptical that explanations in the discussion address the results directly. Specifically, the Authors discuss recently reported results of hippocampal forward sweeps to explain why there would be changes in hippocampal activity when moving from nodes of low degree to nodes of high degree. The authors state that "hippocampal demands might be higher when more paths need to be considered", which would suggest that moving from a node with degree 2 to one with degree 3 should be associated with more hippocampal activity than

moving from a node with degree 6 to one with degree 5, but the data from the study suggest the opposite. In other words, the idea of hippocampal activity scaling with the number of possible paths, to which the authors subscribe, suggests that raw centrality measures should be the measures that correlate with hippocampal activity and not Δs . We suggest that the authors consider a more compelling explanation for the relationship between hippocampal activity and the Δ degree centrality (and not raw centrality).

- Tables S2 and S3 show that many brain regions were found outside of the hippocampus that appear to track these centrality measures as much as, and in many cases, more so than does the hippocampus. These regions are not commented on anywhere in the paper. I would suggest that the Authors provide a comment on the potential role of (at least some of) these extrahippocampal regions. It seems unlikely that the hippocampus is the only brain region that is involved in simulating future navigational experiences, so perhaps there is something of interest here that the Authors may take the opportunity to highlight.

Minor issues:

- The term "categorical difference" used throughout the supplemental tables is used to refer to an ordinal change in degree centrality. This is inconsistent with terms used in the text (i.e. categorical change). Please consider changing wording to be more consistent. Additionally, it may be helpful to put an explanation directly into the text as to why Δ degree centrality was made to be either -1, 0, or 1 instead of how it would be as calculated directly.

- Main text, P6: The Authors should clarify what they mean by "It is possible that prolonged environmental exposure might induce more global processing". The idea of different parts of the hippocampus supporting more local and global environmental processing (i.e. the functional specialization along the antero-posterior axis) is one that, should the authors choose to engage with it, would require further exploration in the discussion.

- Suppl Text: In the description of the MRI acquisitions, please report in-plane voxel size (and not just slice thickness) and matrix size.

- Suppl Text: The description of closeness centrality is unclear. The Authors' initial definition is "how topologically far any two nodes are". However, closeness centrality takes into account all other nodes in the graph. Please revise the wording.

- Suppl Text: The caption for Table 2 is confusing. Specifically, the sentences that refer to Δ param, Δ [param], and *param terms in the table. Please consider re-wording the table, caption, or both to clarify what is meant by each of these terms.

- Suppl Text: Were effects in the hippocampus ROI assessed using a small-volume correction in SPM? If so, please mention this specifically.

- Suppl Tables: Tables S8 and S9 do not add much to the paper and should be removed.

Reviewer #2 (Remarks to the Author):

This impressive study analyzes brain activity during virtual navigation of the streets of London, with route decisions either forced or made by subjects. The authors calculated graph theoretic metrics of degree, closeness, and betweenness, on the London street maps. They found that a region of posterior hippocampus was correlated only with degree centrality, and only during navigation controlled by subjects. This result suggests that the hippocampus is playing out possible futures to be using for decision making, with more activity reflecting more possibilities.

The study is very impressive in the design, the thoroughness of analyses, and the implications of the results. I have only one major concern, outlined below.

Major concern:

An important claim of the paper is that hippocampal activity tracks with degree centrality but not with closeness or betweenness. Hippocampus activity does not come out as significant in these other metrics (according to Table S2), but there is no illustration of how strong hippocampal effects are for the other metrics or whether centrality is significantly stronger than the other metrics. Some assessment of the three metrics in the same hippocampal voxels would be very useful. This could be accomplished by looking at each of the three effects in a full right hippocampus ROI, in a posterior right hippocampus ROI, or by assessing the effects for closeness and betweenness in the voxels that came out for the centrality analysis. All three effects should be displayed in a main text figure in order to give the reader a sense of the relative magnitude of hippocampal activity corresponding to each of the three graph metrics.

Minor points:

1. There should be more discussion of why anterior hippocampus comes out in Table S3 for closeness centrality. Even though anterior hippocampus did not come out as significant when entered into the model by itself, it seems important to acknowledge this finding, especially because it appeared for both navigation and the navigation > con contrast.
2. There should also be more discussion of the potential meaning of the involvement of the other areas listed in Tables S2/3.
3. A few related papers on the hippocampus and sequential/graph structure came to mind that the authors may want to consider mentioning:

Hippocampal involvement in sequential uncertainty/predictability of the future:

Strange, Duggis, Penny, Dolan, & Friston, 2005, Neural networks

Harrison, Duggins, & Friston, 2006, Neural networks

Hippocampal sensitivity to graph community structure in an observational setting:

Reviewer #3 (Remarks to the Author):

Javadi et al in their manuscript entitled "Hippocampal retrieval of network topology to simulate the future" investigate how the hippocampus mediates navigation through city streets using fMRI whole brain imaging and virtual presentation of video clips recorded from the SOHO district in London. They take a unique approach by analyzing SOHO's streets using graph theory, taking measures of degree, closeness, and betweenness for each traveled path. These measures then become the subject of investigation by including them as regressors which seek to explain the fMRI BOLD signal within the right posterior hippocampus. The behavioral paradigm was well thought out with naive subjects first being introduced to pictures of the relevant landmarks for the task, then a tour of the SOHO district which avoided the streets that would be used in the fMRI acquisition portion of the task. After each training portion of the task, subjects were tested to assure they had learned the layout of the environment. The scanned portion then consisted of presentation of images and clips that aimed to simulate navigating the SOHO district. After subjects were oriented in the environment, they were presented with goal locations, and then decision points at intersections where they chose which direction to proceed in order to reach the goal location. Because subjects had not traveled the particular paths presented in the fMRI portion, the task emphasized flexible way-finding in a familiar environment, a type of planning which is thought to be hippocampally dependent. The authors find that the right posterior hippocampus is significantly correlated with the change in degree centrality at the moment when subjects entered new street segments - a time that post-scan interviews revealed as important for path planning.

This is a neat study that attempts to use graph theory to better rationalize about how we represent large-scale environments. The idea is exciting because graph theory provides a nice computational framework for reasoning about how we navigate cities and how we combine both egocentric and allocentric spatial representations to navigate through space. The manuscript is well written, and I believe the study is appropriate for a journal like nature communications. However, I am confused about the reasoning behind their analysis decisions and their consideration of the cognitive aspects of their task. The justification for the design of their analysis are presented only in a short paragraph in the discussion. They justify their design decisions by referencing rodent pre-play literature, where place cells appear to fire in a sequence that represents future trajectories. However, they use a contrast in their analysis that instead considers differences between the present and past (see below), which would appear to be more consistent with re-play rather than pre-play. Additionally, they support their use of a contrast using a rather dated review paper on hippocampal novelty that pertains to learning, not active navigation - the authors ensured that environments were well learned by the time subjects performed the scanned portion of the task so this seems an odd choice of justification. Their final citation in this paragraph appears to be a poster presentation that is not publically available. A solid foundation for the analysis design decisions are therefore lacking. I provide more discussion and recommendations for additional analysis below.

Major Concerns:

Why were only differences between graph metrics of the current segment and the previous segment included as regressors? The place cell finding regarding over-representation of connections between spaces would seem to indicate that activity at the street entry events should reflect the local degree centrality- not a difference between the current and previous degree centrality. The control analyses also follow this pattern where differences between the current and previous metric are used (such as number of visible connecting streets) rather than just the metric at the current position. The authors do not present data to disprove much simpler hypotheses that hippocampal activation simply reflects the local degree centrality (or number of visible connecting streets in the case of the control analyses). To be clear, the presented data are compelling but I think showing that simpler explanations do not explain comparable levels of variance in the data would make for a stronger paper and provide a better justification of the use of graph theory. Therefore, I recommend additional analyses that model the pure metrics rather than contrasts, including control analyses, in order to clarify what is driving hippocampal activation and how subjects might have used a graph representation of the environment to way-find.

Based on the framing of the manuscript and the cognitive task subjects performed (i.e. path planning/simulating the future), I would have expected the authors to focus more on how future paths might determine hippocampal activity rather than using a contrast between the current path and the most recent path. I doubt that this contrast is a good estimate for the planning demands associated with the task. I would like to point the authors to the concept of the "breadth-first search" in graph theory, which is a method for searching a graph in which each of the possible paths is examined before proceeding to a deeper layer in the graph. It appears that this search strategy might be a natural parallel to how subjects perform the task. In the task, subjects are stopped before each node and are asked which of the edges to take. Therefore, answering would appear to require "searching" the next level in the graph and making a selection. Therefore, a measure of the cognitive load that would better reflect computational demands might be created by considering graph metrics associated with all the "n" streets associated with the next intersection (for instance a sum of the graph measures associated with the next layer in the graph, $\sum_{i=1}^n [C_c(p-i)]$). The authors could use these modified metrics as regressors in their analysis in order to better support the manuscript's framing on planning, and to root their analysis in a computational model of how subjects performed the task.

Minor concerns:

Figure 1 panel a is beautiful, but figure S1 does a better job of introducing the concepts in your paper. I suggest swapping them.

Please clarify whether the regressor for street entry points is convolved with a canonical HRF. The methods section states that all regressors are convolved with the canonical HRF, but also states that the street entry points were modeled with zero duration, which is not consistent with a convolution.

Response to Reviewers' Comments

We thank the reviewers for their excellent comments and suggestions. We are grateful for the opportunity to improve our manuscript and resubmit.

Overall, we have substantially updated our revised manuscript with new analyses, results and discussion. Our manuscript is now formatted as an Article and the Methods now appear after the Discussion. We have incorporated section headings to format it for an Article. We have now included previous Supplementary Fig. S1 as main Fig. 1, previous Supplementary Fig. S3 as a new main Fig. 4 and added a new Fig. 5 on prefrontal responses at Detour events.

For new analysis we have:

1. Characterised the response in the anterior hippocampus and included a new Supplementary Fig. S4.
2. Compared posterior hippocampal responses to each of the centrality measures and plotted the mean response for each measure for a posterior hippocampus ROI in our main Fig. 3 and Supplementary Fig. S3.
3. Examined the response of the prefrontal cortex and hippocampus with a measure of the planning demands in of breadth-first search. We find lateral prefrontal regions, but not hippocampus, correlated with this planning demand at Detours where participants report re-planning (new Fig. 5).

Below we respond to issues raised point by point. Text incorporated into the manuscript is shown in red font in this document.

Reviewer #1 (Remarks to the Author):

In this manuscript, the authors present evidence that the right posterior hippocampus codes for the topological connectedness of recently-entered path segments while navigating through an environment. Participants received extensive training in the Soho neighbourhood of London by taking a two-hour tour of the area and learning 23 goal locations. An analysis of Soho's street network provided graph-theoretic measures that were used to test whether brain regions were sensitive to topological properties of the environment. The next day, participants underwent functional MRI while viewing video clips of movement toward a goal along a route composed of street segments and junctions in Soho. The authors found that hippocampal activity increased when participants transitioned from a street segment with fewer connecting segments to one with more connecting segments; they also found that hippocampal activity decreased when transitioning from a segment with more to one with fewer connections. This effect was found to be robust when taking into account visual environmental properties, suggesting that participants retrieved environmental topology from memory when entering a new space.

I believe that this manuscript represents an important step forward in the way that researchers should approach investigation of the neural mechanisms underlying spatial memory and navigation in humans. Particularly, with this manuscript the authors demonstrate that in order to fully understand the neural mechanisms involved in human spatial navigation, it is crucial to take into account environmental properties in addition to individual differences and task demands. However, the Authors should address the issues highlighted hereafter before this manuscript can be considered for publication in Nature Communications.

Major issues:

Rev1 pt1) - While I can appreciate that this manuscript presents novel and interesting findings, the Authors do not convincingly sell the findings in the discussion. It is unclear how these data are likely to change how we think about the role of the hippocampus in the context of navigation and spatial memory. The Authors use the title to suggest that the hippocampus is simulating the future when entering into new spaces, and they provide supporting references for this idea in the

introduction, but fail to incorporate this idea into the discussion. I would suggest re-framing the concluding remarks to both generally highlight the novelty and importance of the findings, and specifically discuss the putative future simulation function that the hippocampus may be demonstrating in the experiment.

Thank you for this advice. We drafted our initial submission as a Brief Communication. In our revision we have expanded our manuscript as an Article to better re-frame the novelty and importance of the findings, where we now refer back to the future simulation references introduced at the beginning of the article and layout how the data provides insight into potential mechanisms (page 11):

“Our observation that posterior hippocampal activity was correlated with the change in degree centrality is consistent with the idea that the hippocampus re-activates representations of paths^{18,20}, with the more paths requiring re-activation the more activity elicited in the hippocampus. Such processing of the local streets is in agreement with the view that the hippocampus helps simulate future possible options to guide choices¹⁻⁵. Hippocampal ‘replay’ or ‘forward-sweeps’^{8,21-23} in the dorsal hippocampus (homologue of the posterior hippocampus) may be the mechanism by which the paths are re-activated; indeed hippocampal replay has been shown to reflect the topology of the environment⁸. The pattern of our data helps clarify between two possible conceptual models of how the hippocampus might process future paths during navigation. Because we found hippocampal activity reflected the change in degree centrality, not the raw degree centrality, a model in which the hippocampus only processes future paths the moment a street is entered is not consistent with our data. Rather, our data agree with a model in which the hippocampus simulates possible paths throughout the journey, with the hippocampal activity we observed reflecting the increase or decrease in the number of potential future paths to be re-activated as each new street is entered.”

Rev1 pt2) - I am skeptical that explanations in the discussion address the results directly. Specifically, the Authors discuss recently reported results of hippocampal forward sweeps to

explain why there would be changes in hippocampal activity when moving from nodes of low degree to nodes of high degree. The authors state that "hippocampal demands might be higher when more paths need to be considered", which would suggest that moving from a node with degree 2 to one with degree 3 should be associated with more hippocampal activity than moving from a node with degree 6 to one with degree 5, but the data from the study suggest the opposite. In other words, the idea of hippocampal activity scaling with the number of possible paths, to which the authors subscribe, suggests that raw centrality measures should be the measures that correlate with hippocampal activity and not Δs . We suggest that the authors consider a more compelling explanation for the relationship between hippocampal activity and the Δ degree centrality (and not raw centrality).

We have now revised our discussion, as noted in the point above, to clarify what the implication is of finding a response related to the change in values rather than raw values, page 11:

"The pattern of our data helps clarify between two possible conceptual models of how the hippocampus might process future paths during navigation. Because we found hippocampal activity reflected the change in degree centrality, not the raw degree centrality, a model in which the hippocampus only processes future paths the moment a street is entered is not consistent with our data. Rather, our data agree with a model in which the hippocampus simulates possible paths throughout the journey, with the hippocampal activity we observed reflecting the increase or decrease in the number of potential future paths to be re-activated as each new street is entered."

Rev1 pt3) - Tables S2 and S3 show that many brain regions were found outside of the hippocampus that appear to track these centrality measures as much as, and in many cases, more so than does the hippocampus. These regions are not commented on anywhere in the paper. I would suggest that the Authors provide a comment on the potential role of (at least some of) these extrahippocampal regions. It seems unlikely that the hippocampus is the only brain region that is involved in simulating future navigational experiences, so perhaps there is something of interest here that the Authors may take the opportunity to highlight.

Thank you for this suggestion. We now make clear on page 10 that no brain regions survive family-wise error correction for whole brain volume but are listed for completeness, as have done in all our prior fMRI research publications:

“Outside our frontal and hippocampal ROIs we found no regions significant when correcting for whole brain volume. For completeness we report all regions active in contrasts at an uncorrected threshold of $p < 0.001$ in supplementary tables.”

We now discuss in more detail the anterior hippocampus and, based on our analysis of planning demands at detours suggested by Reviewer 3, we now report and discuss the role of the prefrontal cortex in simulating future navigation experiences. This is motivated on page 4 at the end of the Introduction:

“While the hippocampus is thought to support retrieval of memory representations to simulate future possibilities, the role of evaluating possible future states for action is argued to be the preserve of the prefrontal cortex. This is based on evidence that damage to the prefrontal cortex specifically impairs planning and problem solving^{9,10}. However, it is not currently clear which regions of the prefrontal cortex evaluate future paths or whether information contained in topological structures is searched to support navigation. We have recently proposed that the lateral frontopolar prefrontal cortex is a suitable candidate region¹⁰. The mechanism by which path evaluation may occur is not known. One potential mechanism is a ‘tree-search’ of all the future branching choices in the network. Consistent with this, recent evidence indicates that humans plan their decisions based on evaluation of each level of the decision tree before proceeding to the next level^{11,12}. For a street network this would involve searching retrieved representations of all the possible paths streets just beyond the next junction. Such a search

mechanism is known as a breadth-first search¹³, which steps through the sequences of possible future choices one level of the decision tree at a time. Prior evidence suggests humans may use this mechanism when planning routes from cartographic maps¹⁴.

Page 6 of the results on anterior hippocampus:

“Right anterior hippocampus tracks the change in the closeness centrality at street entry but not selectively

We found that activity in the right anterior hippocampus was significantly correlated with the change in closeness centrality at Street Entry events during navigation routes, but not during control routes, and was significantly more correlated with the change in closeness during navigation routes than control routes (**Supplementary Fig. S4** and **Supplementary Table S3**). This was the case only when the changes in all three centrality measures were entered into the analysis. We did not find evidence that the anterior hippocampus was more correlated with the change in closeness centrality than the change in degree centrality or the change in betweenness centrality (**Supplementary Table S3**).

We considered that the hippocampal responses to the changes in centrality measures might be driven by visual properties of the environment rather than purely by centrality measures. Thus, we measured various visual properties of the environment which have been examined in prior studies examining graph-theoretic measurements of urban networks¹⁹: line of sight, street width, topological distance to edge of Soho, number of visible connecting streets, visible junctions, and presence of shops, people, or vehicles (**Methods** and **Supplementary Tables S5-**

S8). We found that while none of our measures were significantly correlated with the change in degree centrality, the line of sight and the step depth to the boundary of Soho were correlated with the change in closeness centrality (**Supplementary Table S6**). Thus, we examined whether the anterior hippocampal response was selective to the change in closeness centrality or driven by these other factors. We found anterior hippocampal activity was not significantly correlated with the change in closeness when step depth to boundary or line of sight were included in the analysis (**Supplementary Fig. S4**), nor was the anterior hippocampus significantly correlated with line of sight or step depth to boundary. Thus, the anterior hippocampal response at Street Entry events appears to reflect a combination of environmental properties which relate to the more global importance of a street, e.g. streets closer to the centre of the network and which have a long line of sight.”

Page 9 results examining the prefrontal cortex:

“Inferior lateral prefrontal activity reflects the planning demands at Detours

To explore whether prefrontal activity was specifically related to planning future paths we examined whether responses were correlated with measures in a breadth-first search planning approach¹³. In these models the planning demands are calculated from the sum of the degree centrality in the future street segments to be travelled through to reach the goal (first level of the search tree), such that the more possible paths in the future streets, the greater the demands on planning (**Fig. 5A** and **Methods**). In our post-scan debriefing (see **Methods**) we found participants reported more planning at Detours than at Decision Points, often reporting

that they had planned their choice before the Decision Point. Consistent with this pattern, and with our theoretical prediction that lateral PFC regions might be responsible, we found bilateral inferior lateral PFC was significantly correlated with our measure of breadth-first search planning demands in the first layer of the street network at Detours, but not at Decision Points, and significantly more correlated with planning demands at Detours than Decision points (**Fig. 5B** and **Supplementary Table S11**). We also found that planning demands did not significantly correlate with prefrontal activity during Detours in control routes, where participants were instructed to select one path and the route continued along a different path. The prefrontal response was significantly more correlated with planning demands in navigation route Detours than control route Detours (**Supplementary Table S11**). We found no significant activity correlated with the planning demands when the first layer (**Fig. 5A**) and the second layer of the network were combined to calculate planning demands, indicating that lateral PFC activity reflects the number of path choices in the street segments immediately beyond the next junction, rather than an extensive search of all streets two choices ahead in the network. We also found no significant activity in the hippocampus correlated with the breadth-first search planning demands whether calculated at the first level of the search or both first and second level of search.”

Discussion on anterior hippocampus on page 12:

“Our data provide some fuel for the debate on the functional differentiation of long-axis of the hippocampus^{25,26}. Observation of a posterior hippocampal response to local spatial properties (degree centrality) and an anterior hippocampal response to more global spatial properties

(closeness centrality, see **Fig. 1**) is consistent with the proposal that the posterior region codes fine-grain detail and the anterior codes global information^{18,30-32}. While the posterior hippocampal response is consistent with a local replay of the path representations, an anterior hippocampal response to the change in closeness centrality may reflect a different mechanism. For example, the anterior hippocampus may integrate information during learning about the transition structure across the street network to aid optimal navigation, e.g. which streets will lead to the centre of the network. Consistent with this, recent evidence indicates that the anterior hippocampus may represent the graph community structure during learning the nature of transitions between a set of arbitrary stimuli³³. Notably, our analysis revealed that the anterior hippocampal response was not selective to closeness centrality, and may represent information more generally about the important streets in the environment, with ‘important’ defined here by how central it is in the network both in relation to all streets (closeness), the edge of the space (step depth to boundary) and what can be seen from that street (line of sight). Our results also have implications for city planning by showing, in line with previous studies^{16,19,34}, that certain visual properties of the environment, especially how far one can see (line of sight), but also street width and presence of people, are related to centrality in the city. Future fMRI research with tailored virtual environments will be useful to understand what properties of the environment drive activity in the anterior hippocampus. Past research for example, indicates that the contextual uncertainty of the environment may be important in eliciting anterior hippocampal responses^{35,36}.“

Discussion on prefrontal cortex on page 13:

“While the hippocampus appears to represent information about changes in topological properties, the lateral prefrontal activity reflected the demands of searching the network of possible future paths when re-planning was required at detours. This is consistent with prefrontal regions playing a role in spatial planning during navigation^{10,38,39}. However, it has not been clear which regions of prefrontal cortex are central to this function. We have previously argued that lateral frontopolar regions may be important¹⁰. This proposal was based in part on the observation of increased lateral frontopolar activation in London taxi drivers during re-planning at detours when navigating a virtual simulation of London⁴⁰. Here, we show that activity in this same region, rather than simply being active at detours, is correlated with the path planning demands. Given that the PFC is thought to be domain general in its processing⁹, it seems likely that the lateral PFC regions we have identified here would be engaged during other tasks that require searching a decision tree.”

Minor issues:

Rev1 pt4) - The term "categorical difference" used throughout the supplemental tables is used to refer to an ordinal change in degree centrality. This is inconsistent with terms used in the text (i.e. categorical change). Please consider changing wording to be more consistent. Additionally, it may be helpful to put an explanation directly into the text as to why Δ degree centrality was made to be either -1, 0, or 1 instead of how it would be as calculated directly.

We have now made sure that categorical change is used throughout. We have now included an explanation for why -1, 0 and 1 were used, see Method section on page 22:

“We examined the categorical change (1, 0 or -1) in degree centrality because the range of variation in the change was highly limited.”

Rev1 pt5) - Main text, P6: The Authors should clarify what they mean by "It is possible that prolonged environmental exposure might induce more global processing". The idea of different parts of the hippocampus supporting more local and global environmental processing (i.e. the functional specialization along the antero-posterior axis) is one that, should the authors choose to engage with it, would require further exploration in the discussion.

We have now expanded our discussion of the results in relation to the long-axis of the hippocampus, see page 12 (as noted in our response to point 3 above):

“Our data provide some fuel for the debate on the functional differentiation of long-axis of the hippocampus^{25,26}. Observation of a posterior hippocampal response to local spatial properties (degree centrality) and an anterior hippocampal response to more global spatial properties (closeness centrality, see **Fig. 1**) is consistent with the proposal that the posterior region codes fine-grain detail and the anterior codes global information^{18,30-32}. While the posterior hippocampal response is consistent with a local replay of the path representations, an anterior hippocampal response to the change in closeness centrality may reflect a different mechanism. For example, the anterior hippocampus may integrate information during learning about the transition structure across the street network to aid optimal navigation, e.g. which streets will lead to the centre of the network. Consistent with this, recent evidence indicates that the anterior hippocampus may represent the graph community structure during learning the nature of transitions between a set of arbitrary stimuli³³. Notably, our analysis revealed that the anterior hippocampal response was not selective to closeness centrality, and may represent information more generally about the important streets in the environment, with ‘important’ defined here by how central it is in the network both in relation to all streets (closeness), the edge of the space (step depth to boundary) and what can be seen from that street (line of sight).”

Rev1 pt6) - Suppl Text: In the description of the MRI acquisitions, please report in-plane voxel

size (and not just slice thickness) and matrix size.

We have now included this in the Methods on page 21:

“In each volume thirty-four oblique axial slices, approximately perpendicular to the hippocampus (64 × 64 × 34 matrix size) and 3 mm thick were acquired (3 × 3 × 3 mm voxel size).”

Rev1 pt7) - Suppl Text: The description of closeness centrality is unclear. The Authors' initial definition is "how topologically far any two nodes are". However, closeness centrality takes into account all other nodes in the graph. Please revise the wording.

Thank you for pointing this out. We now describe closeness centrality for street segments as (page 18):

“closeness centrality is the reciprocal of the sum of the topological distance from that segment to all other segments.”

Because we have expanded our manuscript to format for an Article, our previous Supplemental Figure S1 is now main Fig. 1. Which will allow readers to gain a better understanding of the centrality measures in the main article.

*Rev1 pt8) - Suppl Text: The caption for Table 2 is confusing. Specifically, the sentences that refer to $\Delta param$, $\Delta[param]$, and $*param$ terms in the table. Please consider re-wording the table, caption, or both to clarify what is meant by each of these terms.*

We have now changed $[\Delta param]$ to $[\Delta POI]$ (parameter of interest) to avoid confusion with $param$ that is used elsewhere and also sentences that refer to $\Delta param$, $\Delta[param]$, and $*param$ terms:

“ $\Delta param$ refers to *change* of value between previous segment and current segment (value at current segment minus value at previous segment). $[\Delta param]$ refers to categorical change of

param with -1 for $\Delta param < 0$, 0 for $\Delta param = 0$ and 1 for $\Delta param > 0$. * *POI* refers to other parameters of interest: Visible Junction, Visible Connecting Street, path distance, Euclidean distance to goal, step depth to goal, step depth to boundary, light of sight, street width, street length, number of visible people, number of visible vehicles, and number of visible shops. † For this model events in which $[\Delta param] = 0$ were excluded. This was conducted as a follow up to our behavioural experiment, see below. BFS = breadth-first search.”

Rev1 pt9) - Suppl Text: Were effects in the hippocampus ROI assessed using a small-volume correction in SPM? If so, please mention this specifically.

We now state this was the case on page 23 of the manuscript:

“For SPM analysis we used the ROI for a small volume correction applying a threshold of $p < 0.05$ family-wise error (FWE).”

Rev1 pt10) - Suppl Tables: Tables S8 and S9 do not add much to the paper and should be removed.

We would prefer to keep these tables as they show which environmental factors correlate with each other and with our centrality measures. This helps readers understand potential cofounds (or lack of them).

Reviewer #2 (Remarks to the Author):

This impressive study analyzes brain activity during virtual navigation of the streets of London, with route decisions either forced or made by subjects. The authors calculated graph theoretic metrics of degree, closeness, and betweenness, on the London street maps. They found that a region of posterior hippocampus was correlated only with degree centrality, and only during navigation controlled by subjects. This result suggests that the hippocampus is playing out possible futures to be using for decision making, with more activity reflecting more possibilities.

The study is very impressive in the design, the thoroughness of analyses, and the implications of the results. I have only one major concern, outlined below.

Major concern:

Rev2 pt1) An important claim of the paper is that hippocampal activity tracks with degree centrality but not with closeness or betweenness. Hippocampus activity does not come out as significant in these other metrics (according to Table S2), but there is no illustration of how strong hippocampal effects are for the other metrics or whether centrality is significantly stronger than the other metrics. Some assessment of the three metrics in the same hippocampal voxels would be very useful. This could be accomplished by looking at each of the three effects in a full right hippocampus ROI, in a posterior right hippocampus ROI, or by assessing the effects for closeness and betweenness in the voxels that came out for the centrality analysis. All three effects should be displayed in a main text figure in order to give the reader a sense of the relative magnitude of hippocampal activity corresponding to each of the three graph metrics.

We are very grateful for this advice and have conducted a number of further analyses to explore this. Using SPM12 we find that the right posterior hippocampus is more correlated with the change in degree centrality relative to the change in closeness centrality and (at a low threshold) relative to the change in betweenness centrality. These are now reported a new Supplementary Fig. S3, and we have updated Supplementary Table S3 with the z-scores and coordinates of regions identified in these

new analyses.

Figure S3. Comparison of correlation of hippocampal activity with degree, closeness and betweenness centrality. (A) Posterior-hippocampus activity for degree centrality > closeness centrality. The activation map is displayed on the mean structural image at a threshold of $p < 0.005$ uncorrected and 5 voxels minimum cluster size. (B) Posterior-hippocampus activity for degree centrality > betweenness centrality. The activation map is displayed on the mean structural image at a threshold of $p < 0.05$ uncorrected and 5 voxels minimum cluster size. See **Supplementary Table S3** for details.

We also used an ROI approach to extract the mean response in the posterior hippocampus. Which we now display in our new main Fig. 3 panel E:

Figure 3. Posterior hippocampal activity is correlated with the change in degree centrality during navigation. (A) Top Left: Degree centrality plotted for each street segment for an example route (see Fig. 2C). Right: Axiometric projection of the buildings in Soho plotted on a map of Soho. Degree centrality of the route is plotted on the map and projected above. Above the route the graph plots the change in degree centrality for each boundary transition and the top graph plots the evoked response in the right posterior hippocampus at each of the individual boundary transitions (1-6). Analysis of this plot was not used for statistical inference (which was carried out within the statistical parametric mapping framework), but is shown to illustrate the analytic approach. (B-C) Right posterior hippocampal activity correlated

significantly with the change in degree centrality for Nav and Nav > Con during Street Entry events. Statistical parametric maps are displayed with threshold $p < 0.005$ uncorrected on the mean structural image. (D) Parameter estimates for mean activity in the right posterior hippocampus ROI for Nav ($t_{23} = 4.24, p = 0.0003$), Con ($t_{23} = 1.17, p = 0.25$) and Nav > Con ($t_{23} = 4.64, p = 0.0001$) comparisons for a model containing categorical change in degree centrality (see **Supplementary Table S2**). (E) Parameter estimates for mean activity in the right posterior hippocampus ROI for Nav > Con condition for a model containing degree centrality ($t_{23} = 2.28, p = 0.03$), betweenness centrality ($t_{23} = 0.53, p = 0.59$) and closeness centrality ($t_{23} = 0.14, p = 0.88$) measures (**Supplementary Table S3** and **Supplementary Fig. S3**). Error bars denote the SEM. See **Supplementary Fig. S4C** for anterior hippocampal ROI mean responses.

We have accordingly updated the main manuscript in the results section, page 6:

“We also found that the posterior hippocampus was significantly more correlated with the change in degree centrality than the change in closeness centrality and, at a lower threshold, more correlated with the change in degree centrality than the change in betweenness centrality (**Fig. 3E** and **Supplementary Fig. S3**). Thus, the right posterior hippocampus appears to track changes in local path options (degree centrality) when new streets are entered and only when navigating.”

Minor points:

Rev2 pt 2. There should be more discussion of why anterior hippocampus comes out in Table S3 for closeness centrality. Even though anterior hippocampus did not come out as significant when entered into the model by itself, it seems important to acknowledge this finding, especially because it appeared for both navigation and the navigation > con contrast.

Converting our manuscript to an Article provided the space to allow us to discuss this finding and explore it in more detail. We are grateful to the reviewer for suggested we do so. We now report these findings on page 6:

“Right anterior hippocampus tracks the change in the closeness centrality at street entry but not selectively

We found that activity in the right anterior hippocampus was significantly correlated with the change in closeness centrality at Street Entry events during navigation routes, but not during control routes, and was significantly more correlated with the change in closeness during navigation routes than control routes (**Supplementary Fig. S4** and **Supplementary Table S3**). This was the case only when the changes in all three centrality measures were entered into the analysis. We did not find evidence that the anterior hippocampus was more correlated with the change in closeness centrality than the change in degree centrality or the change in betweenness centrality (**Supplementary Table S3**).

We considered that the hippocampal responses to the changes in centrality measures might be driven by visual properties of the environment rather than purely by centrality measures. Thus, we measured various visual properties of the environment which have been examined in prior

studies examining graph-theoretic measurements of urban networks¹⁹: line of sight, street width, topological distance to edge of Soho, number of visible connecting streets, visible junctions, and presence of shops, people, or vehicles (**Methods** and **Supplementary Tables S5-S8**). We found that while none of our measures were significantly correlated with the change in degree centrality, the line of sight and the step depth to the boundary of Soho were correlated with the change in closeness centrality (**Supplementary Table S6**). Thus, we examined whether the anterior hippocampal response was selective to the change in closeness centrality or driven by these other factors. We found anterior hippocampal activity was not significantly correlated with the change in closeness when step depth to boundary or line of sight were included in the analysis (**Supplementary Fig. S4**), nor was the anterior hippocampus significantly correlated with line of sight or step depth to boundary. Thus, the anterior hippocampal response at Street Entry events appears to reflect a combination of environmental properties which relate to the more global importance of a street, e.g. streets closer to the centre of the network and which have a long line of sight.”

New Supplementary Figure 4:

Figure S4. Right anterior hippocampal activity is correlated with the change in closeness centrality. (A) For Navigation routes ($p < 0.05$ family-wise error (FWE) corrected for a priori regions of interest (ROI)), (B) Navigation > Control routes at Street Entry events in a model containing three parametric modulators of the categorical change in degree centrality, closeness centrality and betweenness centrality ($p < 0.05$ FWE for ROI). The activation maps are displayed on the mean structural image at a threshold of $p < 0.005$ uncorrected and 5 voxels minimum cluster size, see **Supplementary Table S3** for details. Follow up control analysis: Because the change in closeness centrality was correlated with the length of the line of sight in

the street segment, and with the step depth to boundary (**Supplementary Table S6**) we included these two parameters in new models to determine if the response shown in A and B was significant when controlling for them. We found no significant hippocampal activity in either model (even at a low threshold of $p < 0.01$ uncorrected), nor was anterior hippocampus significantly correlated with the line of sight, or with step depth to boundary ($p < 0.05$ corrected). (C) Parameter estimates for mean activity in the right anterior hippocampus ROI for Navigation > Control condition for a model containing degree centrality ($t_{23} = 0.05$, $p = 0.95$), betweenness centrality ($t_{23} = 0.28$, $p = 0.77$) and closeness centrality ($t_{23} = 2.51$, $p = 0.01$) measures (* = significant at $p < 0.05$, see **Supplementary Table S3** for details). Error bars denote the SEM. Note: contrasts between these measures (e.g. change in degree centrality > change in closeness centrality) were conducted in the SPM framework - no significant activations were observed in the anterior hippocampus for these contrasts, see **Supplementary Table S3**.

We discuss these findings on page 12:

“Our data provide some fuel for the debate on the functional differentiation of long-axis of the hippocampus^{25,26}. Observation of a posterior hippocampal response to local spatial properties (degree centrality) and an anterior hippocampal response to more global spatial properties (closeness centrality, see **Fig. 1**) is consistent with the proposal that the posterior region codes fine-grain detail and the anterior codes global information^{18,30-32}. While the posterior hippocampal response is consistent with a local replay of the path representations, an anterior hippocampal response to the change in closeness centrality may reflect a different mechanism. For example, the anterior hippocampus may integrate information during learning about the transition structure across the street network to aid optimal navigation, e.g. which streets will lead to the centre of the network. Consistent with this, recent evidence indicates that the anterior hippocampus may represent the graph community structure during learning the

nature of transitions between a set of arbitrary stimuli³³. Notably, our analysis revealed that the anterior hippocampal response was not selective to closeness centrality, and may represent information more generally about the important streets in the environment, with ‘important’ defined here by how central it is in the network both in relation to all streets (closeness), the edge of the space (step depth to boundary) and what can be seen from that street (line of sight). Our results also have implications for city planning by showing, in line with previous studies^{16,19,34}, that certain visual properties of the environment, especially how far one can see (line of sight), but also street width and presence of people, are related to centrality in the city. Future fMRI research with tailored virtual environments will be useful to understand what properties of the environment drive activity in the anterior hippocampus. Past research for example, indicates that the contextual uncertainty of the environment may be important in eliciting anterior hippocampal responses^{35,36}.”

Rev2 pt3). There should also be more discussion of the potential meaning of the involvement of the other areas listed in Tables S2/3.

We now make clear at the end of our results section page 10 that none of these other regions were significant when correcting for whole brain volume and are provided for completeness, as we have done in prior fMRI research publications:

“Outside our frontal and hippocampal ROIs we found no regions significant when correcting for whole brain volume. For completeness we report all regions active in contrasts at an uncorrected threshold of $p < 0.001$ in supplementary tables.”

We now discuss the anterior hippocampus and prefrontal cortex for which we found

significant responses with small volume corrected thresholds. See response to Reviewer 1 point 3 (Rev1 pt3) on page 5 of this document.

Rev2 pt4). A few related papers on the hippocampus and sequential/graph structure came to mind that the authors may want to consider mentioning:

Hippocampal involvement in sequential uncertainty/predictability of the future:

Strange, Duggis, Penny, Dolan, & Friston, 2005, Neural networks

Harrison, Duggins, & Friston, 2006, Neural networks

Hippocampal sensitivity to graph community structure in an observational setting:

Schapiro, Turk-Browne, Norman, & Botvinick, 2016, Hippocampus

We are very grateful for the reviewer bringing these references to our attention, and now cite them in the manuscript on page 12 we cite Schapiro et al., 2016:

“Consistent with this, recent evidence indicates that the anterior hippocampus may represent the graph community structure during learning the nature of transitions between a set of arbitrary stimuli³³.”

On page 13 we cite Strange et al 2005 and Harrison et al. 2006:

“Past research for example, indicates that the contextual uncertainty of the environment may be important in eliciting anterior hippocampal responses^{35,36}.”

Reviewer #3 (Remarks to the Author):

Javadi et al in their manuscript entitled "Hippocampal retrieval of network topology to simulate the future" investigate how the hippocampus mediates navigation through city streets using fMRI whole brain imaging and virtual presentation of video clips recorded from the SOHO district in

London. They take a unique approach by analyzing SOHO's streets using graph theory, taking measures of degree, closeness, and betweenness for each traveled path. These measures then become the subject of investigation by including them as regressors which seek to explain the fMRI BOLD signal within the right posterior hippocampus. The behavioral paradigm was well thought out with naive subjects first being introduced to pictures of the relevant landmarks for the task, then a tour of the SOHO district which avoided the streets that would be used in the fMRI acquisition portion of the task. After each training portion of the task, subjects were tested to assure they had learned the layout of the environment. The scanned portion then consisted of presentation of images and clips that aimed to simulate navigating the SOHO district. After subjects were oriented in the environment, they were presented with goal locations, and then decision points at intersections where they chose which direction to proceed in order to reach the goal location. Because subjects had not traveled the particular paths presented in the fMRI portion, the task emphasized flexible way-finding in a familiar environment, a type of planning which is thought to be hippocampally dependent. The authors find that the right posterior hippocampus is significantly correlated with the change in degree centrality at the moment when subjects entered new street segments - a time that post-scan interviews revealed as important for path planning.

This is a neat study that attempts to use graph theory to better rationalize about how we represent large-scale environments. The idea is exciting because graph theory provides a nice computational framework for reasoning about how we navigate cities and how we combine both egocentric and allocentric spatial representations to navigate through space. The manuscript is well written, and I believe the study is appropriate for a journal like nature communications.

However, I am confused about the reasoning behind their analysis decisions and their consideration of the cognitive aspects of their task. The justification for the design of their analysis are presented only in a short paragraph in the discussion. They justify their design decisions by referencing rodent pre-play literature, where place cells appear to fire in a sequence that represents future trajectories. However, they use a contrast in their analysis that instead considers differences between the present and past (see below), which would appear to be more consistent with re-play rather than pre-play. Additionally, they support their use of a

contrast using a rather dated review paper on hippocampal novelty that pertains to learning, not active navigation - the authors ensured that environments were well learned by the time subjects performed the scanned portion of the task so this seems an odd choice of justification. Their final citation in this paragraph appears to be a poster presentation that is not publically available. A solid foundation for the analysis design decisions are therefore lacking. I provide more discussion and recommendations for additional analysis below.

We are grateful for this advice. By expanding our manuscript to an Article we have taken more space to explain the rationale for our analysis and discussion of our results. We have removed reference to the article examining novelty responses, see page 5:

“We interrogated the fMRI data with three graph-theoretic centrality measures of the street segments: degree, closeness and betweenness. For an explanation of the measures see **Fig. 1, Supplementary Fig. S1** and **Supplementary Table S1**. In previous research we have found hippocampal activity correlated with both raw spatial metrics (e.g. distance to the goal) and the change in metrics (e.g. the change in distance to the goal)¹⁸. Thus we tested whether the hippocampal processing demands might reflect the future simulation demands purely at street entry (raw values) or the change in demands that occurs at street entry (change in values).”

In our discussion on page 11 we have expanded our discussion to explain what the pattern of results may mean:

“Our observation that posterior hippocampal activity was correlated with the change in degree centrality is consistent with the idea that the hippocampus re-activates representations of paths^{18,20}, with the more paths requiring re-activation the more activity elicited in the hippocampus. Such processing of the local streets is in agreement with the view that the

hippocampus helps simulate future possible options to guide choices¹⁻⁵. Hippocampal ‘replay’ or ‘forward-sweeps’^{8,21-23} in the dorsal hippocampus (homologue of the posterior hippocampus) may be the mechanism by which the paths are re-activated; indeed hippocampal replay has been shown to reflect the topology of the environment⁸. The pattern of our data helps clarify between two possible conceptual models of how the hippocampus might process future paths during navigation. Because we found hippocampal activity reflected the change in degree centrality, not the raw degree centrality, a model in which the hippocampus only processes future paths the moment a street is entered is not consistent with our data. Rather, our data agree with a model in which the hippocampus simulates possible paths throughout the journey, with the hippocampal activity we observed reflecting the increase or decrease in the number of potential future paths to be re-activated as each new street is entered.”

The poster presentation on segmentation of space for planning has now been accepted for publication in NIPS, and we have duly updated our reference section.

Major Concerns:

Rev3 pt1). Why were only differences between graph metrics of the current segment and the previous segment included as regressors? The place cell finding regarding over-representation of connections between spaces would seem to indicate that activity at the street entry events should reflect the local degree centrality- not a difference between the current and previous degree centrality. The control analyses also follow this pattern where differences between the current and previous metric are used (such as number of visible connecting streets) rather than just the metric at the current position. The authors do not present data to disprove much simpler

hypotheses that hippocampal activation simply reflects the local degree centrality (or number of visible connecting streets in the case of the control analyses). To be clear, the presented data are compelling but I think showing that simpler explanations do not explain comparable levels of variance in the data would make for a stronger paper and provide a better justification of the use of graph theory. Therefore, I recommend additional analyses that model the pure metrics rather than contrasts, including control analyses, in order to clarify what is driving hippocampal activation and how subjects might have used a graph representation of the environment to way-find.

We examined both raw and the change in values. We now made this much clearer in our new manuscript, see response to the previous point.

Rev3 pt2) Based on the framing of the manuscript and the cognitive task subjects performed (i.e. path planning/simulating the future), I would have expected the authors to focus more on how future paths might determine hippocampal activity rather than using a contrast between the current path and the most recent path. I doubt that this contrast is a good estimate for the planning demands associated with the task. I would like to point the authors to the concept of the "breadth-first search" in graph theory, which is a method for searching a graph in which each of the possible paths is examined before proceeding to a deeper layer in the graph. It appears that this search strategy might be a natural parallel to how subjects perform the task. In the task, subjects are stopped before each node and are asked which of the edges to take. Therefore, answering would appear to require "searching" the next level in the graph and making a selection. Therefore, a measure of the cognitive load that would better reflect computational demands might be created by considering graph metrics associated with all the "n" streets associated with the next intersection (for instance a sum of the graph measures associated with the next layer in the graph, $\sum_{i=1}^n [C_c(p-n)]$). The authors could use these modified metrics as regressors in their analysis in order to better support the manuscript's framing on planning, and to root their analysis in a computational model of how subjects performed the task.

We are very grateful for this suggestion. We previously focused on manuscript purely on

responses to the centrality measures. By incorporating an analysis of demands in a breadth-first search we have made a new discovery in our data. In a recent review we argued that lateral anterior frontal regions might be responsible for tree-search of future paths at detours (Spiers and Gilbert 2015 *Frontiers in Human Neuroscience*, see Fig. 4). Thus, we created a frontal ROI to search for responses in this specific region of the brain, using the methods suggested in the comment above to calculate planning demands. We found bilateral anterior lateral frontal activity scaled with the planning demands of a breadth-first search at Detours during navigation, a response which was absent for Detours in our control routes. In post-scan debriefing we found participants regularly reported re-planning at Detours, but not planning at Decision Points, often stating they had often made their choice before reaching the Decision Point. We found this mirrored our new analysis. We found no evidence for frontal activity correlated with planning demands at Decision Points and significantly more activity correlated with planning demands at Detours. Thus, we are very grateful to reviewer for suggesting we explore this possibility. We now display these new results in a new Figure 5:

Figure 5. Inferior lateral prefrontal activity correlates with the demands of a breadth-first search at Detours. (A) Diagrams of an example street network contrasting scenarios of lower and higher demand breadth-first search. Breadth-first search assumes the search space (street

segments) as a tree and considers all possible solutions within one level before proceeding to the subsequent level. In these diagrams, covering the first layer of the search, the lower demand scenario shows less possible paths while the higher demand scenario shows a greater number of possible paths. For details see **Methods**. (B) The statistical parametric map showing correlation ($p < 0.05$ corrected) of the left and right lateral PFC with planning demands for the first layer of the decision tree (Nav > Con). We found bilateral lateral PFC activity with correlated with planning demands at $p < 0.001$ uncorrected during Detours in navigation routes, but not in control routes. We found no significant correlations when the planning demands. The statistical parametric maps are displayed on the mean structural image at a threshold of $p < 0.005$ uncorrected and 5 voxels minimum cluster size. See **Supplementary Table S11** for details of activations. Comparison of parameter estimates of peak voxel in the right lateral PFC showed a significantly greater response at Detours compared with Decision Points ($t_{23} = 3.49, p = 0.002$).

We now motivate this research question in our introduction on page 4:

“While the hippocampus is thought to support retrieval of memory representations to simulate future possibilities, the role of evaluating possible future states for action is argued to be the preserve of the prefrontal cortex. This is based on evidence that damage to the prefrontal cortex specifically impairs planning and problem solving^{9,10}. However, it is not currently clear which regions of the prefrontal cortex evaluate future paths or whether information contained in topological structures is searched to support navigation. We have recently proposed that the

lateral frontopolar prefrontal cortex is a suitable candidate region¹⁰. The mechanism by which path evaluation may occur is not known. One potential mechanism is a 'tree-search' of all the future branching choices in the network. Consistent with this, recent evidence indicates that humans plan their decisions based on evaluation of each level of the decision tree before proceeding to the next level^{11,12}. For a street network this would involve searching retrieved representations of all the possible paths streets just beyond the next junction. Such a search mechanism is known as a breadth-first search¹³, which steps through the sequences of possible future choices one level of the decision tree at a time. Prior evidence suggests humans may use this mechanism when planning routes from cartographic maps¹⁴."

We now describe the methods on page 24:

"To analyse a measure of how search might occur as opposed to just detecting the future possibilities, we calculated the demands in a breadth-first search (BFS) in graph theory, which is a method for searching a graph¹³. We ran two levels of search with (1) sum of the degree centrality measures of all street segments connecting to the next immediate junction (see Fig. 5A) and (2) the combined sum of the sum of the degree centrality measures of all street segments connecting to the next immediate junction and the sum of degree centrality measures for all streets segments connecting to the subsequent junctions on the optimal path to the goal. BFS assumes calculation based on the degree centrality, but we considered whether the search demands might change if calculated with closeness or betweenness centrality. We found that BFS demand measures using degree centrality, closeness centrality or betweenness centrality were highly correlated ($r > 0.8$), and resulted in nearly identical SPM

results to those from BFS using degree centrality. To test whether lateral prefrontal cortex was involved in this search we created a lateral frontal ROI that encompassed the regions predicted in our recent review¹⁰ using the bilateral inferior and mid lateral frontal ROIs from the WFU_PickAtlas⁶¹.”

Results on page 9:

“Inferior lateral prefrontal activity reflects the planning demands at Detours

To explore whether prefrontal activity was specifically related to planning future paths we examined whether responses were correlated with measures in a breadth-first search planning approach¹³. In these models the planning demands are calculated from the sum of the degree centrality in the future street segments to be travelled through to reach the goal (first level of the search tree), such that the more possible paths in the future streets, the greater the demands on planning (**Fig. 5A** and **Methods**). In our post-scan debriefing (see **Methods**) we found participants reported more planning at Detours than at Decision Points, often reporting that they had planned their choice before the Decision Point. Consistent with this pattern, and with our theoretical prediction that lateral PFC regions might be responsible, we found bilateral inferior lateral PFC was significantly correlated with our measure of breadth-first search planning demands in the first layer of the street network at Detours, but not at Decision Points, and significantly more correlated with planning demands at Detours than Decision points (**Fig. 5B** and **Supplementary Table S11**). We also found that planning demands did not significantly correlate with prefrontal activity during Detours in control routes, where participants were instructed to select one path and the route continued along a different path. The prefrontal

response was significantly more correlated with planning demands in navigation route Detours than control route Detours (**Supplementary Table S11**). We found no significant activity correlated with the planning demands when the first layer (**Fig. 5A**) and the second layer of the network were combined to calculate planning demands, indicating that lateral PFC activity reflects the number of path choices in the street segments immediately beyond the next junction, rather than an extensive search of all streets two choices ahead in the network. We also found no significant activity in the hippocampus correlated with the breadth-first search planning demands whether calculated at the first level of the search or both first and second level of search.”

Discussion on page 13:

“While the hippocampus appears to represent information about changes in topological properties, the lateral prefrontal activity reflected the demands of searching the network of possible future paths when re-planning was required at detours. This is consistent with prefrontal regions playing a role in spatial planning during navigation^{10,38,39}. However, it has not been clear which regions of prefrontal cortex are central to this function. We have previously argued that lateral frontopolar regions may be important¹⁰. This proposal was based in part on the observation of increased lateral frontopolar activation in London taxi drivers during re-planning at detours when navigating a virtual simulation of London⁴⁰. Here, we show that activity in this same region, rather than simply being active at detours, is correlated with the path planning demands. Given that the PFC is thought to be domain general in its processing⁹, it

seems likely that the lateral PFC regions we have identified here would be engaged during other tasks that require searching a decision tree.”

Minor concerns:

Rev3 pt3) Figure 1 panel a is beautiful, but figure S1 does a better job of introducing the concepts in your paper. I suggest swapping them.

We thank the reviewer for making this suggestion, and have now included Supplementary Fig. S1 as main Fig. 1. Previous main Fig. 1 is now present at main Fig. 2.

Rev3 pt4) Please clarify whether the regressor for street entry points is convolved with a canonical HRF. The methods section states that all regressors are convolved with the canonical HRF, but also states that the street entry points were modeled with zero duration, which is not consistent with a convolution.

Yes, it is. We have taken the standard approach of modelling events used in SPM12, which is to treat them as onsets with zero duration. This stick function is then convolved with the HRF. See e.g. Josephs, O., Turner, R., & Friston, K. (1997). Event-related fMRI. *Human Brain Mapping*, 5(4), 243-248. In our methods section on page 22 we state:

”For effects with duration zero we took the standard approach of modelling events used in SPM12. This stick function is then convolved with the HRF⁵⁴.”

REVIEWERS' COMMENTS:

Reviewer #1 (Remarks to the Author):

The Authors have revised the manuscript substantially, and they have addressed all my issues appropriately. The manuscript is now, in my opinion, ready to be published in Nature Communications. It will make a significant contribution to the field.

Reviewer #2 (Remarks to the Author):

The authors have thoroughly addressed my comments. I enthusiastically endorse publication.

Reviewer #3 (Remarks to the Author):

The authors have addressed all of my major concerns and I approve of the revised manuscript. I recommend that the authors revise figure 5's caption due to grammatical errors.